# Attractive serial dependence arises during decision-making

**Jiangang Shan** [1]*, **Jasper E. Hajonides** [2,3], **Nicholas E. Myers** [4]

**1** Department of Psychology, University of Wisconsin-Madison, Madison, Wisconsin, United States of America, **2** Oxford Centre for Human Brain Activity, Wellcome Centre for Integrative Neuroimaging, University of Oxford, Oxford, United Kingdom, **3** Department of Experimental Psychology, University of Oxford, Oxford, United Kingdom, **4** School of Psychology, University of Nottingham, Nottingham, United Kingdom

* jiangangshancogneuro@gmail.com

## Abstract

Recall of stimuli is biased by stimulus history, variously manifested as an attractive bias toward or repulsive bias from previous stimuli (i.e., serial dependence). It is unclear when attractive versus repulsive biases arise and if they share neural mechanisms. A recent model of attractive serial dependence proposes a two-stage process in which adaptation causes a repulsive bias during encoding that is later counteracted by an attractive bias at the decision-making stage in a Bayesian-inference-like manner. Neural evidence exists for a repulsive bias at encoding, but evidence for the attractive bias during the response period has been more elusive. We recently (Hajonides et al., J Neurosci 43:2730–40, 2023) showed in a working memory task that while different stimuli in trial history exerted different (attractive or repulsive) serial biases on behavioral reports, during encoding the neural representation of the current item was always repulsively biased. Here, we assessed whether this discrepancy between neural and behavioral effects is resolved during subsequent decision-making. Multivariate decoding of human magnetoencephalography data during working memory recall showed a neural distinction between attractive and repulsive biases that is consistent with the two-stage model: an attractive neural bias was found in recall period. And stimuli that created a repulsive bias on behavior led to an early repulsive neural bias that is likely to have already been incorporated during the encoding phase. The neural attractive bias late in the trial was replicated in an independent electroencephalogram experiment. Our results suggest that attractive (but not repulsive) serial dependence arises during decision-making, and that priors that influence post-perceptual decision-making are updated by the previous trial's target, but not by other stimuli.

**Data availability statement:** Data supporting main findings of the study are available at osf. io/xd7rh/.

**Funding:** This research was funded by National Institutes of Health (https://www.nih.gov/; grant number MH131678) and the Wellcome Trust (https://wellcome.org/; Awards 092760/Z/10/Z and 201409/Z/16/Z/ to N.E.M). The Wellcome Center for Integrative Neuroimaging (supporting N.E.M. and J.E. H.) is supported by core funding from the Wellcome Trust (Grant 203139/Z/16/Z). The funders did not play any role in the study design, data collection and analysis, decision to publish, or preparation of the manuscript.

**Competing interests:** The authors have declared that no competing interests exist.

**Abbreviations:** EEG, electroencephalogram; MEG, magnetoencephalography.

## Introduction

Our sensory apparatus contends with the dynamic and noisy nature of sensory inputs by taking advantage of the apparent stability over time of natural environments [1]. For example, in working memory tasks participants' recall in the current trial is biased toward the memorandum presented in the last trial. This ubiquitous effect is known as serial dependence and has been shown with different types of tasks and memoranda, from simple features like numerosity [2], orientation [3–5], or position [6, 7], to complex features like facial expressions [8, 9] and 3D motion [10].

Serial dependence can manifest as either an attractive bias toward (e.g., [3, 4]) or a repulsive bias from (e.g., [6, 11]) the previous stimulus. Both attractive and repulsive biases can emerge in the same task (e.g., [12, 13, 14]). A line of behavioral studies have argued that serial dependence (attractive or repulsive bias) happens in a two-stage process: The repulsive bias arises from efficient encoding or sensory adaptation during the encoding of the current stimulus [15]. During this stage, the limited sensory resource is allocated efficiently via sensory adaptation towards recently encountered features to increase sensitivity for the likeliest stimuli [16, 17]. As a result of this, the new stimulus is biased repulsively from the previous stimulus [18, 19]. Attractive bias, by contrast, is believed to arise at a post-perceptual decision-making stage [6], where participants use Bayesian inference to inform the current report ([5, 13, 20]; but see [21, 22]). Based on the implicit assumption that the state of the world in the recent past is predictive of the state of the world now, trial history is incorporated into a prior. This prior information is combined with the current evidence and leads to the final report being attracted toward the previous stimulus [5, 6, 20], resulting in temporally smoothed and more stable perception (or memory) of the world [23].

Neural evidence is vital to testing models of serial dependence as it can track the emergence of biased representations across all stages of a trial. Although we have shown evidence for a repulsive neural bias during encoding ([12]; see also Sheehan and Serences [24]), studies directly looking into the neural bias in the decision-making stage are still lacking. Moreover, in our previous study [12], different items in trial history induced different behavioral effects: Participants' report in the current trial was attracted toward the cued item from the previous trial (henceforth "previous target"), while the report of the second sample was, at the same time, repulsively biased away from the first sample in the same trial ("sample 1"). However, during encoding the neural representation of the current stimulus was repulsively biased away from both sample 1 and the previous target. This neural effect at the encoding stage is contrary to the prediction of the continuity field theory of serial dependence, in which the attractive bias arises during perception or encoding into working memory of the new stimulus [3], but is consistent with the two-stage model [15]. Thus, we speculated that the discrepancy between the neural effect and behavioral report is resolved in the subsequent decision-making stage, as predicted by the two-stage model. In the current study, we re-analyzed the data from this study but focused on the memory recall period. We tested two hypotheses in our analyses: First, the neural evidence of attractive bias should arise during the post-perceptual decision-making stage.

Second, this attractive bias should be context-dependent—namely that only the target from the previous trial, but not the other stimulus from the current trial, led to an attractive neural bias.

## Results

We re-analyzed the data from [12] to investigate the neural correlates of attractive serial dependence. Human participants completed a visual working memory task while their neural activity was recorded with magnetoencephalography (MEG). On each trial (Fig 1A) two gratings were presented sequentially and participants were cued to recall the orientation of one of them. Two different serial biases were found: Participants' reports were attracted toward the cued item from the previous trial ("previous target", Fig 1B; $t(19) = 5.8241$, $p < 0.0001$). By contrast, when the second sample was cued it was repulsively biased away from the first sample on the same trial ("sample 1", $t(19) = -2.4394$, $p = 0.0247$, see [12] for a detailed assessment of behavioral serial biases for each condition). For reference, we also examined the bias induced by sample 2 on the report of sample 1 in trials where both items were presented and sample 1 was cued. There was no significant sample-2-on-sample-1 bias ($t(19) = 0.683$, $p = 0.5029$). Please note, however, this bias should not be considered a serial dependence effect in the sense considered here as sample 1 was encoded before sample 2 and any influence would have been retrospective.

To assess whether there was any relationship between the participant's behavioral serial bias and their performance in the task, we calculated the Pearson correlation between participants' previous-target-induced bias or their sample-1-induced bias with their mean absolute recall error (in all trials or in both-shown-sample-2-cued trials, for two types of bias respectively). We observed a significant correlation between the mean absolute recall error and the previous-target-induced bias (Pearson's $r = 0.6465$, $p = 0.0021$), such that the stronger the attractive serial bias, the worse the recall performance. This suggests the more a participant relied on the previous target, the worse their performance is. For the sample-1-induced bias, we did not find a significant correlation (Pearson's $r = 0.3009$, $p = 0.1973$).

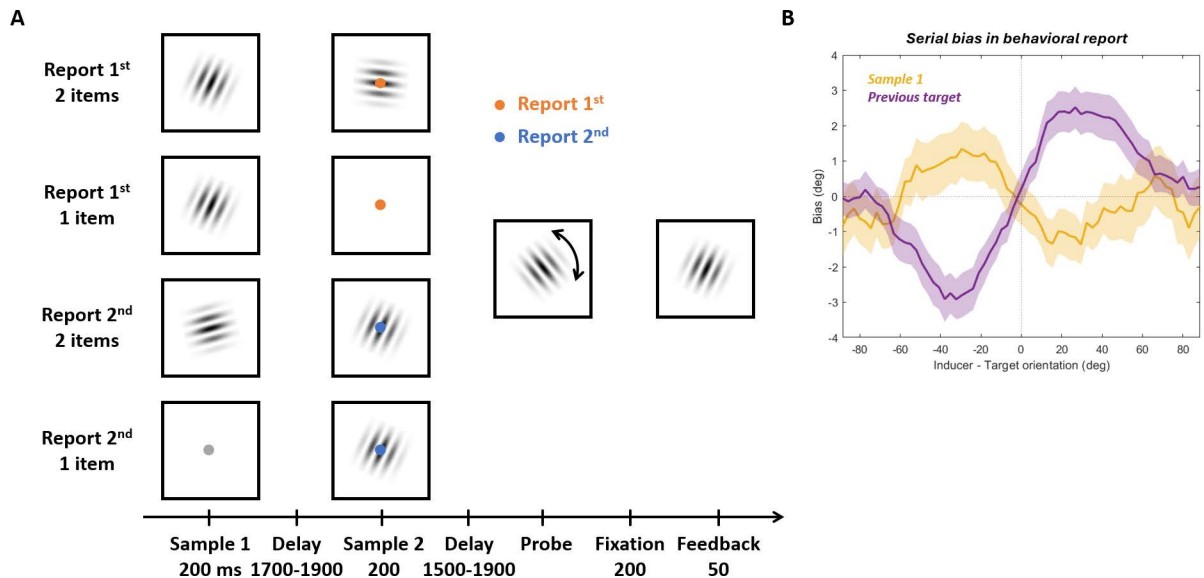

**Fig 1. Task design and behavioral results. (A)** Two arrays were presented sequentially in each trial. In half the trials, two gratings were presented serially (sample 1 and sample 2). In the other half of trials, a single grating was presented either at the first (25% of trials) or second (25% of trials) time point. A color cue presented at the same time as the second sample indicated which grating the participant should recall. When the probe appeared, participants rotated it to match the orientation of the target item using button presses. The report was registered by pressing a third button. A grating with the correct orientation was presented as feedback. **(B)** Behavioral serial dependence induced by sample 1 or previous target. The mean signed error of recall errors as a function of the relative orientation of the inducer to the current target. Data was smoothed for visualization purpose only. The smoothing was done by binning the relative orientation of the inducer into 64 evenly spaced, overlapping bins, with each bin containing the 25% of trials closest to its bin center. Shading indicates SEM.

We previously found [12] that the neural biases during the encoding of stimuli failed to fully explain the behavioral biases: in the encoding phase the neural representation of the stimulus was repulsively biased away from sample 1, mirroring the repulsive behavioral bias, but also biased away from the previous target, in contrast to the attractive bias in behavior. Here, we re-analyzed the data from this study but focused on a previously unexamined part of the trial: the memory recall period.

To examine the neural bias in the recall phase we first identified time periods during which the target orientation could be decoded. Multivariate decoding of the target orientation was applied to the MEG data time-locked to the probe onset (see Methods for details). We held out each trial in turn and calculated its neural similarity (using negative Mahalanobis distances) to the average MEG response to all possible stimulus orientations (split into 10 evenly spaced orientation bins, with the held-out trial's orientation in the center), yielding a tuning curve consisting of the neural similarity to each orientation bin. An unbiased representation of the target should show a peak in the bins closest to 0°. To quantify decoding quality, responses in each bin were transformed to vectors (pointing toward the bin's orientation and length equal to its similarity to the test trial) and then averaged to a mean vector that was projected onto the 0° vector (Fig 2A). The target orientation could be decoded from 254 to 834 ms after probe onset (Fig 2A left; two-tailed cluster-based permutation test, $p = 0.00004$). We separately time-locked the data to the end of recall (i.e., when participants submitted their report by pressing a button). Target decoding was significant from 163 to 0 ms before response (Fig 2A right; $p < 0.00001$).

We next tested whether the representation of the current target was biased by previous stimuli by examining whether the neural tuning curve was shifted away from 0°. We took neural similarity to clockwise ([−72°, −18°]) and counterclockwise orientation bins ([18°, 72°]) as evidence for clockwise versus counterclockwise biases, respectively. The difference between them was calculated as the asymmetry index. A positive asymmetry index indicates a clockwise neural bias.

To investigate the bias induced by the previous target, we compared the asymmetry index of the current target for the trials where the previous target was clockwise versus counterclockwise to the current target. First, we analyzed the average bias in the time window of significant target decoding (254–834 ms after probe onset; Fig 2A). We did not find any bias of the neural representation of the current target caused by the previous target (Fig 2B left; $t = 0.4816$, $p = 0.6356$). However, a bias seemed to arise later in the recall period (Fig 2B left, starting from ~1,000 ms). As decision-making is a dynamic and variable process (response time distribution in Fig 2A), we investigated the bias locked to the end of the recall period, before participants reached their decisions. In the last 2,000 ms before participants completed the recall, the previous target attractively biased the current target's neural representation (Fig 2B right and 2D; 2,000–0 ms before response, $t(19) = 3.7624$, $p = 0.0013$). This attractive bias was still marginally significant when we restricted the analysis to the narrow time window of 163 ms with significant target decoding (163–0 ms before response, $t(19) = 2.0861$, $p = 0.0507$).

To investigate whether sample 1 biased the target during the recall period, we focused on trials where both samples were presented and the 2nd sample was cued to recall. In contrast to the effect of the previous target, sample 1 caused a repulsive bias on the current target's neural representation when it was reactivated by the probe (254–834 ms; $t(19) = −3.5613$, $p = 0.0021$; Fig 2C and 2E). This paralleled its repulsive effect on recall. For the response-locked analysis, in contrast to the previous-target-induced effects reported above, we found no bias induced by sample 1 (last 2,000 ms before response: $t(19) = −1.0003$, $p = 0.3298$; last 163 ms before response: $t(19) = 0.3203$, $p = 0.7523$).

To further investigate the origins of the attractive and repulsive biases, we conducted a searchlight decoding analysis. The sample-1-induced repulsive bias was most prominent in posterior sensors (Fig 3A). This topography closely resembles the inverse topography of the reactivated target decoding (Fig 3B; Pearson rho = −0.5973, $p < 0.0001$). For the attractive bias found in the response-locked analysis, posterior sensors showed the largest bias (Fig 3C), again similar to the

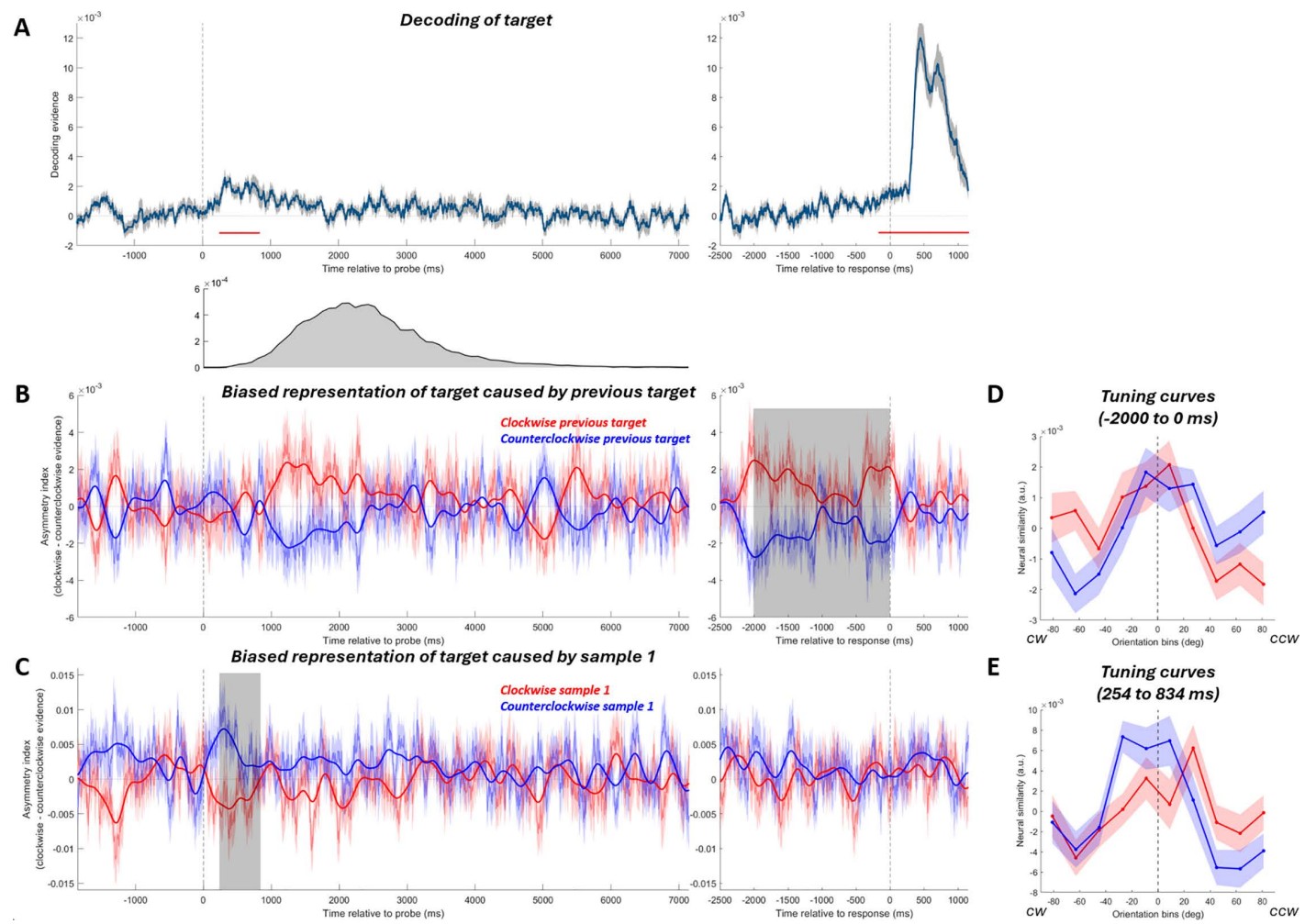

**Fig 2. Decoding of the current target and the neural bias on the target induced by sample 1 or the previous target. (A)** Orientation decoding of the current target. Decoding evidence of zero would suggest no successful decoding. Red horizontal lines indicate timepoints with significant decoding assessed with cluster-based permutation tests. The strong decoding of target after the response was likely driven by the feedback stimulus showing the correct orientation of the target, which was presented 200 ms after the response. Gray shading indicates SEM. The plot below shows the distribution of response times (estimated with the time when participants submitted their reports) across participants and trials. Ninety-nine percent of the reports were made 700–6,200 ms after probe onset. Note that in all probe-locked analyses, the later part of the epoch (while and after most responses were given, starting at approximately 2,000 ms) *contained increasing amounts of noise from irrelevant signals after the current recall period*. **(B)** Bias of the neural representation of the current target caused by the previous target. Trials were sorted into two groups according to whether its previous target is clockwise (red line) or counterclockwise (blue line) to the current target. The y-axis shows the mean asymmetry index, with a positive number indicating a clockwise-biased neural representation. A smoothed time course is also shown on top (bold lines) for visualization purpose only. The smoothing was done by convolving a Gaussian filter (s.d. = 80 ms) with the raw asymmetry index time course (thin lines). Shadings around the thin lines indicate SEM. The gray-shaded regions indicate time windows with which paired *t*-tests were conducted. **(C)** Bias of the neural representation of the current target caused by sample 1, with the same plotting conventions as in **B. (D)** Tuning curves (mean-centered within each participant) showing the multivariate reconstruction of the current target, in clockwise-previous-target trials (red) and counterclockwise-previous-target trials (blue) separately, averaged in the last 2 s before response. Shadings indicate SEM. CCW, counterclockwise; CW, clockwise. **(E)** Tuning curves showing the reconstruction of the current target, in clockwise-sample-1 trials (red) and counterclockwise-sample-1 trials (blue) separately, averaged in the time window from 254 to 834 ms after probe onset.

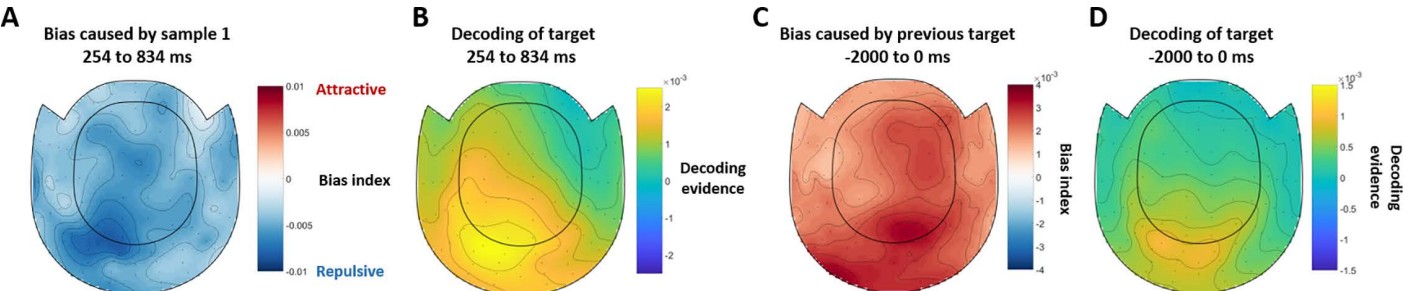

**Fig 3. The MEG topographies showing which sensors contributed more to the effects. (A)** The topography of the sample-1-induced repulsive bias, averaged in the time window from 254 to 834 ms after probe onset. The bias index was calculated by subtracting the asymmetry index of counterclockwise inducer trials from clockwise inducer trials. A positive bias index represents an attractive neural bias. **(B)** The topography showing which sensors contributed the most to the decoding of target in the same time window. **(C)** The topography of the previous-target-induced attractive bias in the last 2 s before response. **(D)** The topography of target decoding in the same time window.

topography of target decoding in the same time window (Fig 3D; rho = 0.668, $p < 0.0001$), although right frontal areas also appeared to show an attractive bias. This suggests that the neural representation of the current target in posterior areas may be biased by the previous target when it is reinstated at the readout stage.

It is possible that the attractive bias we found could be merely driven by the visual response to the probe on the screen. That is, since the participant's report was attracted toward the previous target and participants rotated the probe stimulus on screen to make this report, it is possible that the attractive bias was actually a sensory effect rather than reflecting a biased memory representation.

To rule out this possibility, we looked directly into the bias of the current target representation driven by the participant's report on each trial. The logic of this analysis was as follows: On a trial-by-trial basis, the serial bias often diverges from the actual report error because the serial bias only accounts for a relatively small proportion of overall response error variance. We re-sorted the same trials from the analysis of previous-target bias, but this time according to whether the reported orientation in the current trial (not the previous target) was clockwise or counterclockwise relative to the target orientation. If the stimulus on the screen biased the representation of the target, we should have seen an attractive neural bias towards the direction of the reported orientation. However, no bias caused by the participant's report survived cluster-based permutation testing. Furthermore, in the time window of the previous-target-induced bias (i.e., 2000–0 ms before response), no significant bias was found (Fig 4B; $t(19) = 0.3208$, $p = 0.7519$). During the last 163 ms right before the response (i.e., when the decoding of target was significant), the bias was also non-significant ($t(19) = 0.1235$, $p = 0.9030$).

Furthermore, by comparing the time course of the previous-target-induced attractive bias and those of the decoding of the participant's report, we again found evidence inconsistent with this possibility (S1 Fig). We conclude that the attractive neural bias towards the previous target was related to biased memory representations and not sensory feedback during recall.

We further investigated whether there was a reactivation of the previous target (c.f., [11, 25]) during the current recall period. The reactivation of the previous target during the current trial has been speculated to play a role in attractive serial dependence [25–27]. Thus, we tested whether the previous target could be successfully decoded during the recall period, especially during the time window where we found the attractive bias. However, in the time window exhibiting the attractive bias no reactivation of the previous target was found (Fig 5, −2,000 to 0 ms relative to the report completion, $t(19) = −0.1872$, $p = 0.8535$). Additionally, no reactivation of the previous target throughout the time course survived cluster-based permutation testing. For completeness, we also tested the reactivation of the previous target in the other two time windows, and no significant decoding was found (254–834 ms after probe onset, $t(19) = 1.5307$, $p = 0.1423$; −163 to 0 ms relative to the report completion, $t(19) = −0.4773$, $p = 0.6386$).

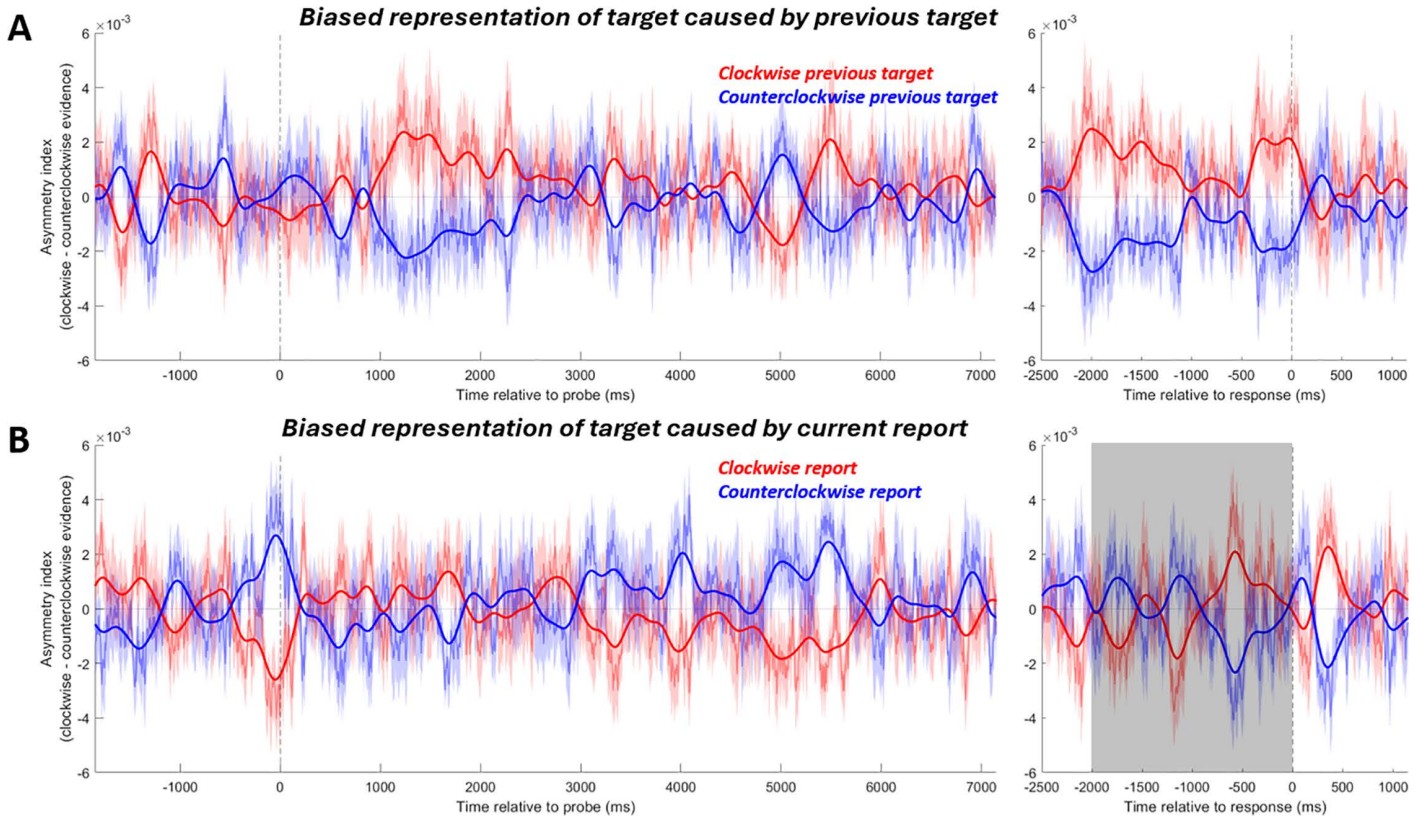

**Fig 4. Asymmetry index of the representation of the current target. (A)** For reference, the neural bias of the target induced by the previous target (identical to Fig 2B). **(B)** When trials are sorted according to the relative orientation of the participant's report in the current trial, there is no bias. Plotting conventions are the same as in Fig 2.

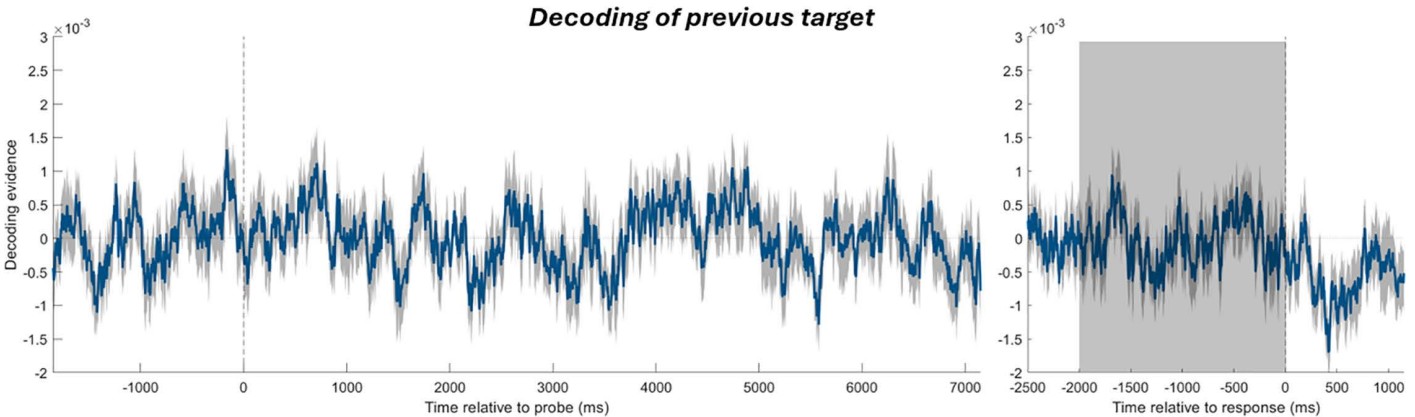

**Fig 5. Decoding of the target from the previous trial, with the same plotting conventions as in Fig 2A.** Since no significant period of decoding was found with cluster-based permutation test, we also applied one-sample *t*-tests in the time window where a previous-target-induced attractive bias was found (gray-shaded region). No successful decoding was found in this time window either.

## Replication in an independent EEG dataset

To establish the robustness of these neural-bias effects, we sought to replicate, in an independent dataset, the recall-period-attractive bias we found in the current MEG experiment and the repulsive bias during the encoding stage reported in the previous study [12].

We analyzed data from an electroencephalogram (EEG) experiment in which 30 human participants performed a visual working memory task (Fig 6A, see Methods for a detailed description of the task) with two items and a retrocue. Each trial was started by participants pressing down the two response keys and holding them down throughout the trial. Participants were required to encode two sequentially presented, randomly oriented gratings in working memory (250 ms each, separated by a 500 ms delay). Then, after another delay of 500 ms, an auditory cue indicated which of the two items would be probed after the final delay of 1,500 ms. The probe stimulus appeared at a random orientation, and participants dialed it to the orientation of the cued memory item. Participants needed to release both keys and then press a key to dial the probe. They then pressed the "Enter" key to submit their report. Fifty ms after the report was made, feedback (two small light gray discs presented at the edge of the probe grating and indicating an imaginary line corresponding to the correct orientation of the cued item) was shown for 200 ms. Masks (random Gaussian noise stimuli presented at a rate of 250 ms per stimulus) were used in the 500 ms before the sample 1 onset, as well as during the three delay periods. We conducted the behavioral and neural analyses on this EEG dataset in the same way as in the MEG dataset (see Methods).

At the behavioral level, we found an attractive serial bias induced by the previous target on the report in the current trial (Fig 6B, $t(29)$ = 4.2902, $p$ = 0.00018). Sample 1 did not induce a significant bias in the report of sample 2 (Fig 6B, $t(29)$ = 0.2677, $p$ = 0.7908). The discrepancy between this result and the repulsive sample-1-induced bias found in the MEG experiment could be due to differences in task design. For instance, the retrocue here was presented after encoding of the sample 2, rather than at the same time. This delay may have changed the encoding strategy to reduce interference. The use of noise masks between samples and in the delay may have additionally affected the EEG task (see Methods for details). For reference, we again examined the bias induced by sample 2 on the report of sample 1. There was a significant sample-2-on-sample-1 attractive bias ($t(19)$ = 6.8377, $p$ < 0.0001). A possible interpretation of this result is that participants occasionally reported sample 2 when sample 1 was cued (a swap error), leading to an apparent attractive effect. As above, this bias should not be considered a serial dependence effect as sample 1 was encoded before sample 2.

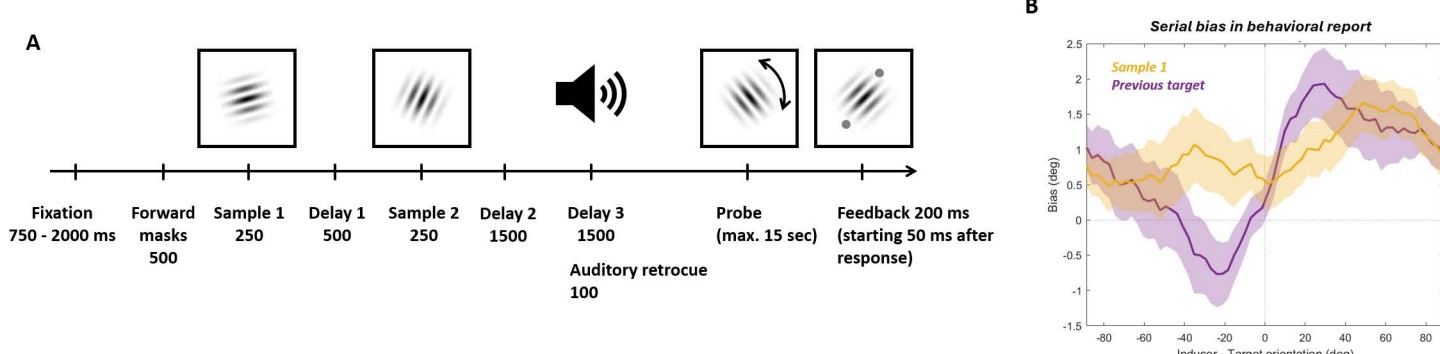

**Fig 6. Task design and behavioral results of the EEG dataset. (A)** Two samples were presented sequentially in each trial. An auditory retrocue indicated which grating the participant should recall. Then, the probe appeared with a random starting orientation, and participants rotated it to match the orientation of the target item using button presses. The report was registered by pressing "Enter". Feedback was provided with two small light gray discs presented at the edge of the grating to indicate an imaginary line corresponding to the correct orientation of the cued item. **(B)** Behavioral serial dependence induced by sample 1 or previous target. The mean signed error of recall errors as a function of the relative orientation of the inducer to the current target. Data was smoothed for visualization purpose only. The smoothing was done by binning the relative orientation of the inducer into 64 evenly spaced, overlapping bins, with each bin containing the 25% of trials closest to its bin center. Shading indicates SEM.

PLOS Biology

We again assessed whether there was any relationship between the participant's behavioral serial bias and their performance in this task. Pearson correlation between participants' previous-target-induced bias and their mean absolute recall error (in all trials) was calculated. Here we found no significant correlation (Pearson's $r = 0.2454$, $p = 0.1911$), although numerically a stronger attractive bias predicted higher absolute recall error (i.e., worse performance).

Before examining neural biases, we established whether the orientation of sample 1, sample 2, and the target could be decoded during the encoding, maintenance, and decision-making phases of the task. Decoding of sample 1 became significant shortly after stimulus onset and persisted throughout the presentation of sample 2 until just before the onset of the retrocue (Fig 7A, 157 to 1361 ms, $p < 0.00001$, cluster-based permutation test). For sample 2, three significant clusters were detected (Fig 7A): after sample 2 onset until approx. 500 ms after retrocue onset (893–1962 ms, $p < 0.00001$), in the post-cue delay (2,488–2,869 ms, $p = 0.0155$), and after probe onset (3,332–3,841 ms, $p = 0.0097$). Decoding of the cued target orientation (Fig 7B) was significant in the post-cue delay (2,313–2,793 ms, $p = 0.0138$) and after probe onset (3,152–5,161 ms, $p < 0.00001$).

We next examined the neural biases in different phases of the trial. During encoding, we found no significant bias induced by the previous target on the neural representation of the current-trial sample 1 or sample 2 (S2A and S2B Fig). While this contrasts with the significant across-trial repulsive neural bias reported in our previous study [12] (see Discussion for potential causes of the discrepancy), we still found no evidence for the emergence of the attractive bias during encoding. Within the trial, we found sample 1 caused a significant repulsive bias on the neural representation of sample 2 when it was encoded into working memory (S2C Fig), which replicated the encoding-stage repulsive neural bias reported in the previous study [12].

Our main aim was to replicate the attractive neural bias during the post-perceptual decision-making stage. To this aim, we examined periods of significant target decoding after the onset of retrocue, after which participants can start recalling the cued item to prepare for making the report. In this experiment, this included a period of significant decoding after retrocue onset but before probe onset (possibly partially due to the visual noise masks driving a constant visual response, which has been shown to increase decoding of working memory contents [28]), and a period of significant decoding after the probe onset. The previous target induced a prolonged attractive bias on the current-target neural representation in this decision-making stage (Fig 8A). We combined the two periods with significant decoding (2,313–2,793 ms, after retrocue onset before probe onset; and 3,152–5,161 ms, in the probe period) into a time window and investigated the previous-target-induced neural bias in it. A significant attractive bias was found (Fig 8A, $t(29) = 2.155$, $p = 0.0396$). For the individual time windows, the attractive neural bias is marginally significant in the probe-period time window (3,152–5,161 ms, $t(29) = 1.9383$, $p = 0.0624$), and it did not reach significance in the other time window (2,313–2,793 ms, after retrocue onset before probe onset, $t(29) = 1.5983$, $p = 0.1208$). Note that the cluster-based permutation test did not find significant clusters of neural bias. Compared to the attractive neural bias in the MEG experiment (which first emerged at ~1,000 ms after probe onset, Fig 2B), the attractive bias in the EEG experiment started earlier in the trial (after retrocue onset, Fig 8A). This difference across the two experiments could be explained by two factors: First, the significant decoding of the current target during the post-retrocue delay before the probe in the EEG dataset (Fig 7B), which is absent in the MEG dataset (Fig 2A); Second, the different temporal profile of the MEG task, in which the retrocue was presented at the same time as sample 2. Thus, the stronger sensory-driven signal of sample 2 could have masked any subtle signal corresponding to the attractive bias and made it impossible to be detected by our analysis.

For the sample-1-induced neural bias during the decision-making stage, we repeated the same analysis but sorted trials according to the relative orientation of sample 1. Mirroring the absence of a behavioral effect, sample-1 did not induce a neural bias here in the combined time window (Fig 8B, 2,313 to 2,793 ms and 3,152–5,161 ms, $t(29) = -1.179$, $p = 0.248$). For the individual time windows, sample-1 did not induce a neural bias in either one of them (3,152–5,161 ms, in the probe period, $t(29) = -1.1015$, $p = 0.2797$; and 2,313–2,793 ms, after retrocue cue onset before probe onset, $t(29) = -0.5126$, $p = 0.6121$). No significant bias was detected by the cluster-based permutation test.

Next, we conducted a control analysis (as in the MEG analysis) to test whether the previous-target-induced attractive bias during the recall period could have been driven by the visual response to the rotated probe on the screen instead of

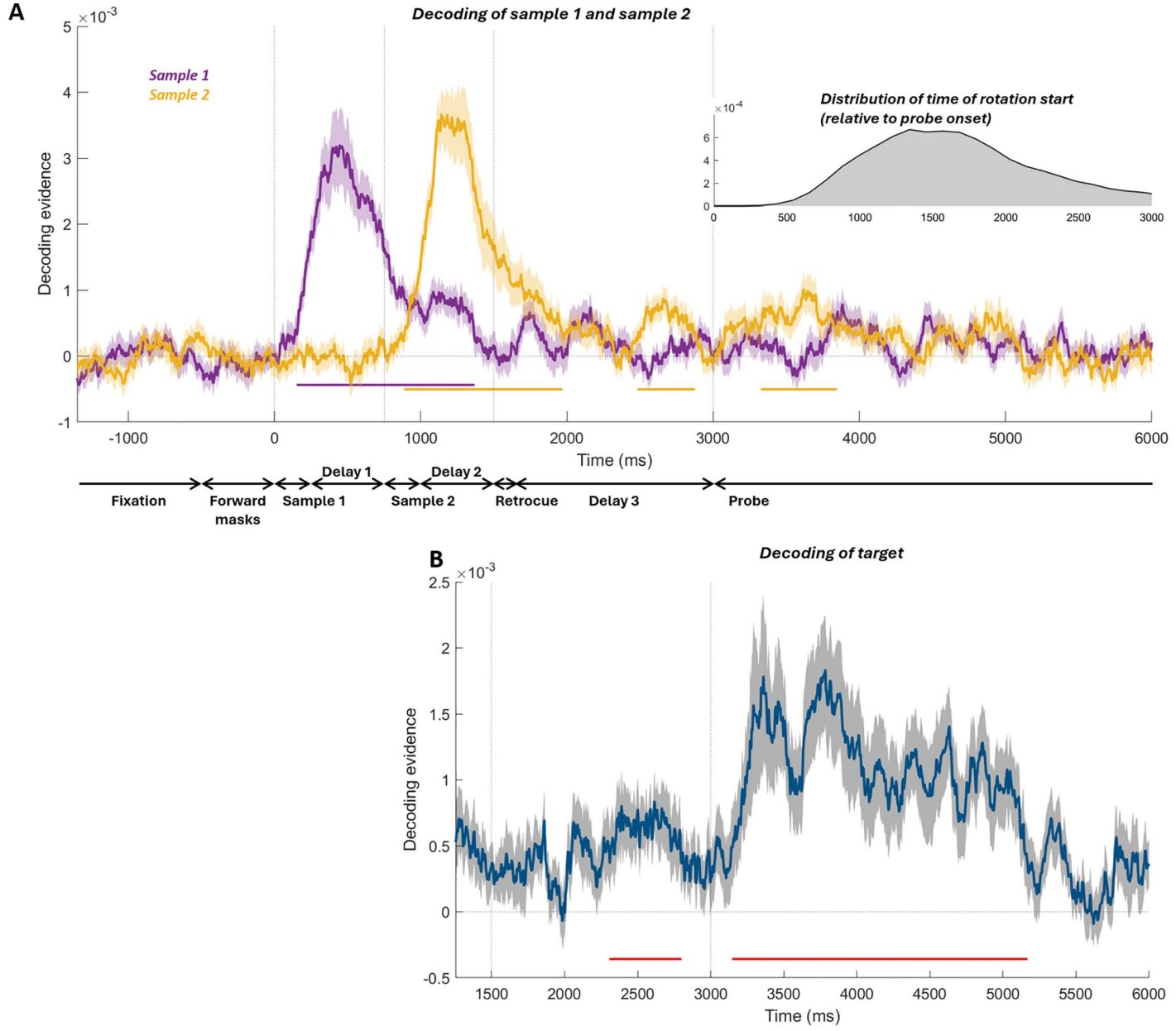

**Fig 7. Decoding of sample 1, sample 2, and target of the current trial. (A)** Orientation decoding of sample 1 (purple curve) and sample 2 (orange curve). Decoding evidence of zero would suggest no successful decoding. Horizontal lines indicate timepoints with significant decoding assessed with cluster-based permutation tests. Shading indicates SEM. The inset plot shows the distribution of the time when participants started rotating the probe (see Methods). Across participants and trials, 99% of rotation-start times fell into the range of 635 to 5,367 ms. 91.70% of rotation-start times were shorter than 3 s. The dotted vertical lines indicate the time of sample 1 onset (0 ms), sample 2 onset (750 ms), retrocue onset (1,500 ms), and probe onset (3,000 ms), respectively. **(B)** Orientation decoding of the target.

reflecting a decision-making process. All trials were sorted according to the reported orientation at the end of the trial (relative to the current target orientation). Different from the results from the MEG analysis, in the EEG dataset we observed an attractive bias in the recall period, with a significant difference detected by the cluster-based permutation test (Fig 9, 4,108 to 4,357 ms, $p = 0.0409$).

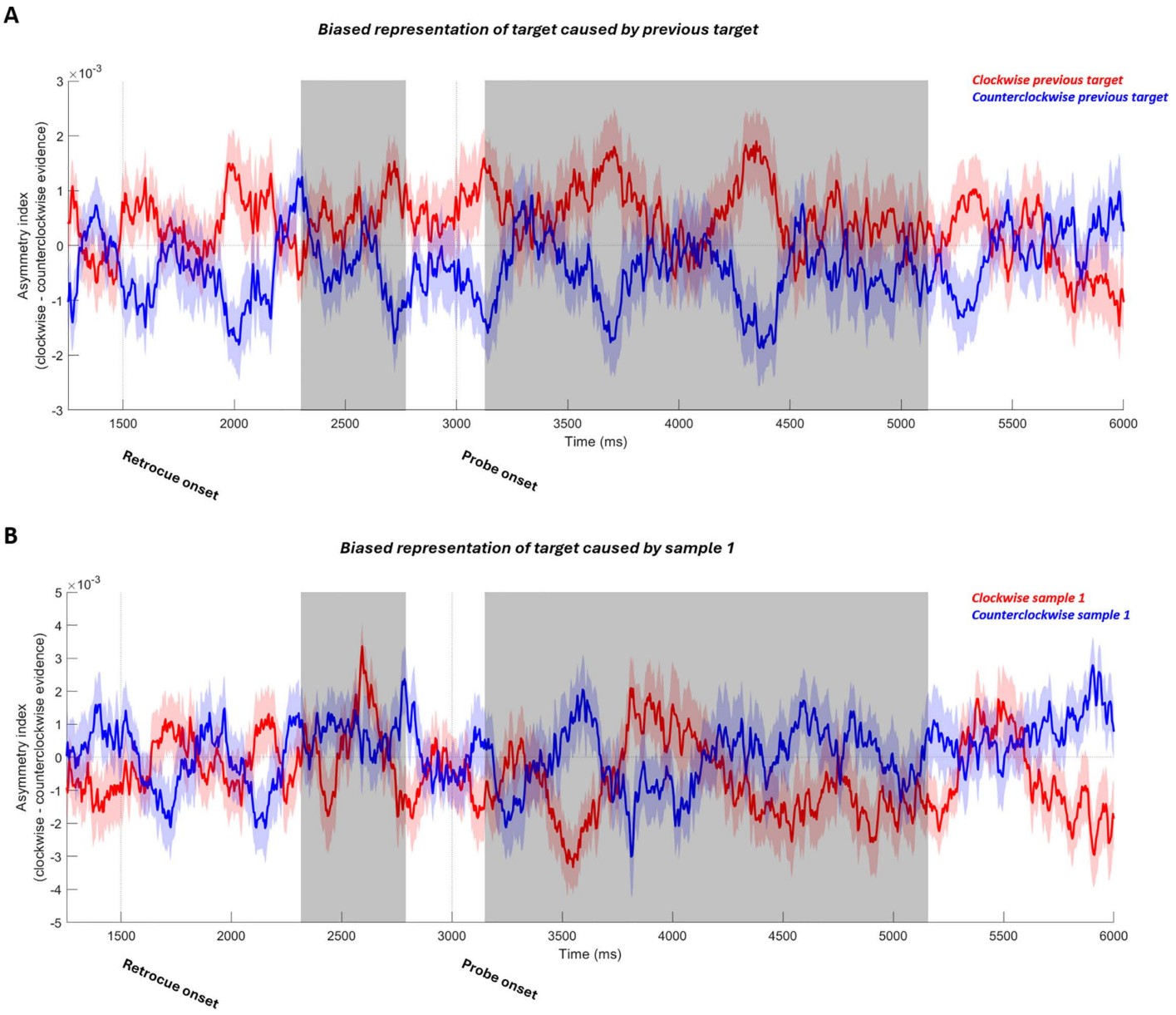

**Fig 8. The neural bias induced by the previous target (first row) and sample 1 (second row) during recall of the current target. (A)** Bias of the neural representation of the current target caused by the previous target. Trials were sorted according to whether the previous target was clockwise (red line) or counterclockwise (blue line) to the current target. The y-axis shows the mean asymmetry index, with a positive number indicating a clockwise-biased neural representation. Shadings around the thin lines indicate SEM. The gray-shaded regions indicate time windows when sample decoding was significant and over which data were averaged for statistical inference. **(B)** Bias of the neural representation of the current target (i.e., sample 2) caused by sample 1 in sample-2-cued trials. Trials were sorted according to the orientation of sample 1 relative to the current target.

To rule out that the previous-target-induced attractive bias was driven by the visual responses to the rotated probe on screen, we repeated the analysis of the previous-target-induced bias but limited the analysis time window on each trial to before participants began rotating the probe. Before participants began responding, the initial orientation of the probe stimulus was independent from the target orientation, and therefore could not have contributed to the attractive bias. The averaged asymmetry index was calculated in the retrocue-onset-to-rotation-start time window as well as the

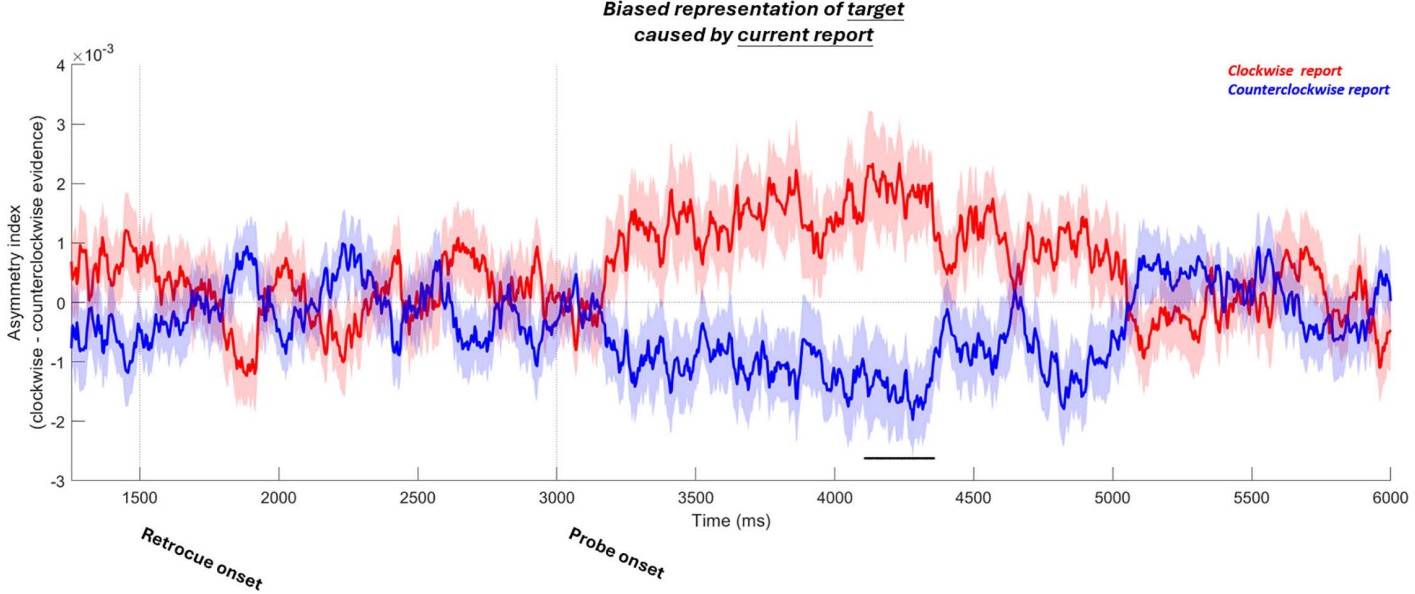

**Fig 9. Asymmetry index of the representation of the current target when trials were sorted according to the relative orientation of the participant's report in the current trial.** Plotting conventions are the same as in Fig 8.

probe-onset-to-rotation-start time window. That is, for each trial, the end of the time window is determined by the time when participants began rotating the probe stimulus in that trial (see the inset of Fig 7A). We found the previous target caused a significant attractive bias in the retrocue-onset-to-rotation-start time window ($t(29) = 2.3526$, $p = 0.0256$). In the much shorter probe-onset-to-rotation-start time window (see Fig 7A inset), the attractive bias trended in the same direction but failed to reach significance ($t(29) = 1.6094$, $p = 0.1184$). These results indicated that there was, indeed, a genuine previous-target-induced attractive bias that is not related to the sensory response to the rotated probe. As above, we did not expect to see any neural bias caused by sample 1. This is indeed what we found (retrocue-onset-to-rotation-start time window: $t(29) = −0.5559$, $p = 0.5825$; probe-onset-to-rotation-start time window: $t(29) = −0.1962$, $p = 0.8458$).

Lastly, we tried to decode the orientation of the previous-trial target during the current trial. As shown in Fig 10, the decoding of the previous target was only significant before the onset of current-trial sample 1 (−1,350 to −494 ms, which roughly corresponds to the intertrial fixation period, $p = 0.00008$; and −359–18 ms, during which the forward masks were on screen, $p = 0.0081$). The previous target could not be decoded in the combined time window in which we tested the previous-target-induced attractive bias (2,313–2,793 ms and 3,152–5,161 ms, $t(29) = 0.9282$, $p = 0.3609$). Nor could it be decoded in the two individual time windows (3,152–5,161 ms, in the probe period, $t(29) = 0.8575$, $p = 0.3982$; and 2,313–2,793 ms, after retrocue onset before probe onset, $t(29) = 0.5043$, $p = 0.6178$). This suggests that the observed attractive neural bias was unlikely to be driven by a reactivation of the previous target during decision-making (Please see Discussion).

## Discussion

In the present study, we examined how distinct attractive and repulsive serial dependence biases caused by a previous target versus a previous stimulus could arise during working memory recall. Consistent with their respective behavioral effects, we found that sample 1 caused a repulsive bias during the time window when the current target was reactivated by the probe, while the bias caused by the previous target was found later in the trial and was attractive

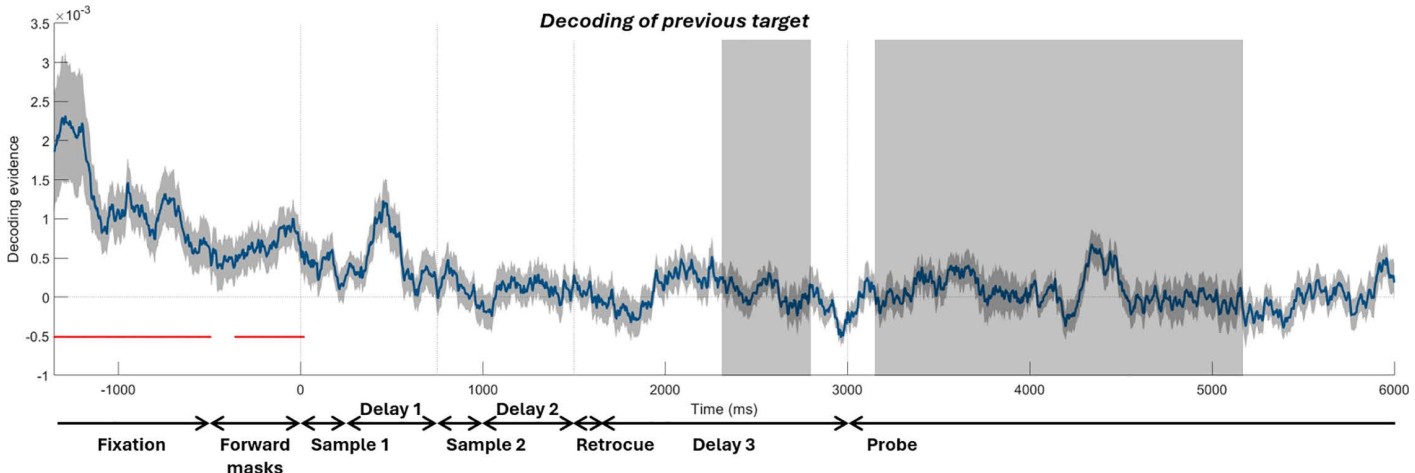

**Fig 10. Decoding of the target from the previous trial, with the same plotting conventions as in** Fig 7. The gray-shaded regions indicate time windows in which one-sample *t*-tests were conducted.

in nature. To test the robustness of these effects we conducted the analyses on an independent EEG dataset. During the recall period, the previous target again attractively biased the neural representation of the current target, mirroring the previous-target-induced attractive bias in participants' behavioral reports. In the EEG dataset we found no sample-1-induced bias in participants' reports and correspondingly found no neural bias in the recall period.

The origin of the attractive serial dependence effect, whether perceptual or post-perceptual, has been a hotly debated question in the field. On the one hand, the continuity field theory of serial dependence proposes that an attractive bias arises during perception and serves to integrate similar stimuli and stabilize noisy perception over time [3]. In support of this view, attractive biases have been observed in early sensory processing [21, 22] and neural activity reflecting the previous target emerges either shortly before [25] or during [29] encoding of a new stimulus. An alternative proposal is that serial dependence of working memory representations develops post-perceptually. For example, a recent model posits that serial bias is the result of a two-stage process: During the encoding of the new stimulus, trial history repulsively biases the encoded representation via sensory adaptation, and during readout of the memory representation, the trial history causes an attractive bias in a Bayesian-inference-like manner [6, 13, 15, 20]. These two hypotheses are not mutually exclusive, as a multiplicity of mechanisms may be creating both attractive and repulsive biases at several processing stages. In addition to neural evidence for a repulsive bias during the encoding of information [12, 24, 30], an attractive neural bias later in the trial has recently been reported [30, 31]. Our results are consistent with this finding and extend it in two important ways: First, instead of showing the alignment between representational codes for the current- and previous-trial information [30], we examined the neural representation of the target itself and showed that this representation was biased toward the previous target in the decision-making stage and, in a second experiment, the late memory delay [31]. More importantly, biases induced by different previous items were examined in the current study. We found results indicating the context-dependent nature of attractive serial bias: only the target from the previous trial, but not the other item from the current trial caused this Bayesian-inference-like attractive bias. While serial dependence is believed to be modulated by attention, such that previously attended information causes a stronger attractive bias [3, 32, 33], our finding suggests different previously attended information may not be incorporated into the prior to the same degree.

While the current results speak for the two-stage models in which the attractive serial bias develops post-perceptually in a working memory task, they do not rule out the possibility that the attractive serial bias can emerge at the perceptual

level, particularly in tasks focusing on perception rather than memory [34]. Many previous studies have found evidence showing that attractive serial dependence can occur at the time of perception [7, 22], or that it influences the early processing of visual information [21, 29]. Further studies are needed to understand when and how the attractive serial bias happens in a perceptual or post-perceptual stage.

The first stage of the two-stage model of serial dependence predicts a repulsive neural bias happening during the encoding of new stimuli. This effect has been shown in [12, 24, 30] but was not fully replicated in our EEG experiment: During the encoding of sample 2, we found that sample 1 repulsively biased the neural representation of this new sample. In contrast to the MEG dataset [12], however, we did not see a repulsive bias induced by the previous target during the current-trial encoding of either sample 1 or sample 2. This discrepancy could be explained by the specific format of the end-of-trial feedback showing the correct orientation of the target (which provided much weaker visual input in the EEG task compared to the MEG task), and the use of forward visual masks in the EEG study (which likely reduced the influence of the sensory adaptation to the previous target; see Methods). Despite the differences between the two experiments, neither experiment finds evidence for an attractive bias at encoding.

During the recall period of the MEG experiment, we found both attractive and repulsive behavioral biases were accompanied by a corresponding neural bias. Specifically, the sample-1-induced repulsive bias happened along with the reactivation of the current target and the previous-target-induced attractive bias emerged later in the recall. The sample-1-induced repulsive neural bias, at first glance, may seem to be at odds with the two-stage model of serial dependence, since the repulsive bias should happen during encoding, but not the post-perceptual decision-making stage. However, the time course (Fig 2A and 2C) and topography (Fig 3A and 3B) of this repulsive bias closely paralleled those of the target, suggesting that it was already biased upon reactivation. Since the representation of the target was already biased repulsively from sample 1 when it was encoded [12], the repulsive bias found at the very beginning of the recall period could simply reflect the sensory bias from the encoding stage instead of emerging anew during decision-making. For the EEG dataset, we again found a recall-period neural bias consistent with participants' behavioral reports. After the retrocue and during recall, the previous target caused an attractive bias on the neural representation of the current target, consistent with the previous-target-induced attractive bias on the behavioral level. Sample 1 here did not induce a repulsive serial bias in behavior, and thus, no significant neural bias was found.

Interestingly, the time course of the previous-target-induced attractive neural bias was different across the two experiments. In the MEG dataset, the attractive bias first emerged at ~1,000 ms after the onset of the probe. In the EEG dataset, the attractive bias appeared to emerge after the retrocue onset. There were a number of task differences that could have led to this discrepancy. First, the target orientation was decodable in the late delay of the EEG experiment, possibly driven by increased signal from the presentation of noise stimuli. In the MEG experiment, there was no target decoding in the delay. Since the attractive bias is contingent on the ability to decode the target, this may have led to the absence of a delay-period bias. One possibility is that the attractive bias gradually emerges throughout the delay period and decision-making stages. This could reconcile the various differences in the results between our two experiments and reports of post-encoding attractive neural biases emerging at different task stages [30,31]. It might also explain why the attractive bias, once detectable in the MEG experiment, appeared largely at posterior sensors (Fig 3C), as it may have already incorporated the bias at this point and reinstated this biased representation in posterior areas. Whether this is indeed the case and, if so, sensory reinstatement is important for the development of the bias in behavior, is a question we cannot answer here. Second, the timing of cue presentation differed between the experiments. In the EEG experiment the cue appeared after the presentation of sample 2. This decoupled encoding of sample 2 from selection of the target item. An attractive bias may emerge soon after selection of the cued item (as in the EEG experiment) only when it is already fully encoded in WM. In the MEG experiment, this was not possible because selection and encoding co-occurred. Additionally, sample 2 evoked a strong sensory signal with its presentation [12], which could had swamped the weaker signal corresponding to the attractive serial bias.

Why did the previous target and sample 1 exert different biases? From the perspective of Bayesian inference, our result suggests that the previous target was incorporated into the prior to inform the decoding of the current target, whereas sample 1 was not. Behavioral studies on serial dependence have demonstrated that the attractive bias depends on many factors, from the subjective perception of the item [21, 27, 35, 36], spatial closeness [33] and shared irrelevant features [37] between the previous and current items, to how participants handle this no-longer-relevant information [14, 38]. In the current study, while participants may have incorporated trial history into the prior by default, for the other memorandum shown in the same trial (i.e., sample 1), they may have had a higher motivation to individuate it and thus excluded it from the prior. Another important difference between the previous target and the sample 1 is that while the participants had recalled the previous target and received the feedback showing the correct orientation of the target after the recall, they did not need to recall sample 1. The recall and feedback may lead to deeper or prolonged processing of the previous target and putatively results in a greater impact on serial dependence. However, we believe it is unlikely that the attractive bias was driven purely by an action plan, since previous studies have shown that a response to the previous target is not necessary to produce serial dependence [3, 7, 8], and in the current tasks (both datasets) the action participants needed to conduct was independent of the actual orientation of the target (see Methods).

The repulsive serial bias in behaviors has been reported in previous studies where a previously encoded item needs to be actively suppressed or actively removed. Rafiei and colleagues [39], for example, studied the serial dependence effect in an odd-one-out visual search task in which participants were required to find a target among distractors. While target orientation recall was attracted toward the target orientation in the previous trial, it was, at the same time, biased away from the mean orientation of the distractor ensemble. The orientation of distractors could be actively suppressed by the participants, and, as a result, cause a repulsive bias on the perception of the current target. Similarly, Shan and Postle [14] showed that while a no-longer-needed item that was removed from working memory by withdrawal of attention induced the classical attractive serial bias on the report in the following trial, an item that was actively removed instead induced a repulsive bias. In the current study, sample 1 became irrelevant when sample 2 was cued. Thus, participants could have used a strategy of active suppression or removal, causing the repulsive bias. However, at the mechanistic level, what happened in the current study may be different from active suppression/removal. Both active suppression and active removal are believed to exert their influence on a following stimulus during the perception/encoding of the new stimulus [39–41]. In contrast to this, in the current MEG experiment, both sample 1 and the previous target caused repulsive neural biases when the new item was presented [12], and the difference between them emerged during decision-making, where the previous target, but not sample 1, induced an attractive neural bias. Similar to this, for the EEG experiment, the sample-1-induced repulsive bias happened when sample 2 was presented, which was before the participants knew which item to report, and thus before they knew which item they could suppress or remove from working memory. Hence, we believe the difference between the handling of sample 1 versus the previous target in both of our experiments happened during the recall but not the encoding stage.

Reactivation of previous information in the current trial has been reported in several studies and is believed to be important for attractive serial dependence. Neural representation of the previous stimulus has been found to be un-decodable during the intertrial interval but decodable again around the onset of the current stimulus [11, 42]. Reactivation of previous information has also been observed in the later part of the delay [27] and during the recall or probe period [30, 31]. The strength of the reactivation may correlate with the magnitude of serial bias [25–27]. In a previous analysis of our MEG dataset, a reactivation of the previous target was found during the encoding of current stimuli [12]. During the probe period, we did not see any reactivation of the previous target in both datasets (Figs 5 and 10), in contrast to [30, 31]. It could be that the weaker signal of the reactivated previous target was swamped by the stronger visual signal evoked by the probe that remained on the screen throughout the probe period and by the visual signal evoked by the noise masks used in the EEG task. While we cannot rule out reactivation that was undetectable to our decoding method, an alternative possibility is that prior information is stored through an activity-silent mechanism that influences decision-making (c.f., [25, 43]) without requiring reactivation. Further studies at the neural level (which, for example,

minimize the physical salience of the probe stimulus and, thus, minimize its influence on the neural activity during recall) are needed to study the possible ways in which the prior exerts its influence.

Taken together, by using multivariate decoding of brain activity during working memory encoding and recall, we demonstrate an encoding-period repulsively biased neural representation of the second sample driven by the first sample, as well as a recall-period attractively biased neural representation of the current target driven by the previous target. The recall-period attractive biases were only induced by the target from the previous trial, but not the other stimulus previously encoded in the same trial, demonstrating that this post-perceptual decision-making process is context-sensitive. The current study provides evidence for a two-stage model of serial dependence in which adaptation during encoding is overcome by reliance on a temporally smoothed prior during recall of working-memory representations.

## Methods

### MEG experiment

We analyzed the dataset from [12]. In this study [12], human volunteers participated and provided their informed written consent according to the procedures approved by the Central University Research Ethics Committee of the University of Oxford (MSD-IDREC-C1-2012-98). The study was conducted in accordance with the Declaration of Helsinki. Twenty participants (11 females) completed a working-memory task where they needed to recall the orientation of one of two gratings (diameter of 6° visual angle, two cycles per degree, 50% contrast, and tapered by a Gaussian envelope with a 1.5° SD) presented in the trial (Fig 1A). Each trial started with a black central fixation for 800 ms, followed by the first grating with a random angle presented at the center of the screen for 200 ms. After an interstimulus interval of 1,700–1900 ms, the second grating with a random angle was shown for 200 ms. The orientations of the two gratings are independent from each other. Simultaneously with the second grating onset, the central fixation changed color to signal if the first grating or the second grating would be probed. After a delay of 1,700–1900 ms, a probe grating was shown on the screen with random starting orientation. Participants were asked to adjust the orientation of the probe to match the orientation of the target grating by pressing two buttons to rotate the probe (Note that this arrangement, along with the random starting orientation of the probe, rendered the action participants needed to conduct to make the report independent from the actual orientation of the target). Responses were submitted by pressing another button. Response times were calculated as the time between the onset of the probe and time of this button press. Then, feedback was provided by showing a grating with the correct orientation. Each trial could consist of 1 or 2 gratings, and the probed item could be either the first or the second grating in 2-grating trials. In trials where the first grating was not shown, the fixation dot changed from black to gray to indicate the omission.

Each participant completed 400 trials in total. In 200 trials, both items were presented, with 100 report-first-item trials and 100 report-second-item trials. In 100 trials, only the first item was presented and probed to be recalled. In 100 trials, only the second item was presented and probed to be recalled. Trials were randomly mixed.

### Serial bias at the behavioral level

Serial dependence in the participant's reports induced by the sample 1 or the previous target was assessed in a model-free way. The averaged signed report error was calculated separately for trials where the inducer is [0°, −45°] or [0°, +45°] relative to the target, and the difference between them is calculated as the index of serial bias for each participant. For the sample-1-induced bias, we focused on trials where both items were presented and the second item was cued to be the target. For the previous-target-induced bias, we used all trials except the first trial in each block, as there was no previous target.

### MEG data preprocessing

To analyze the MEG activity in the probe period, the data was downsampled to 250 Hz and re-epoched around the probe onset (from 2 s before probe onset to 13 s after probe onset). Epochs were visually inspected to remove trials

containing big jumps in activity or non-stereotyped artifacts. Then, an independent component analysis was run on the remaining trials and components related to blinks and cardiac activities were visually identified and removed from the data.

## Multivariate decoding based on Mahalanobis distance

We train the decoder on all non-rejected trials to decode the current target orientation. 367.85 ± 25.609 (mean ± s.d.) trials for each participant were included in this decoding analysis. Response times across all trials ranged from 0.50 to 12.63 s, and 99% of reports were made 0.70 to 6.20 s after probe onset (Fig 2A bottom). Thus, we only considered MEG data from 2 s before probe onset to 7 s after probe onset in this analysis. For each time point, data from all 306 sensors with a sliding time window of 37 time points (i.e., 148 ms) were combined into a vector with 11,322 elements. To reduce its dimensionality, principal component analysis was applied to a matrix of data where each row corresponds to the vector from one trial, maintaining 90% of the variance. Across participants, 155.55 ± 22.53 and 163.65 ± 18.79 components were retained in the probe-locked and response-locked analyses, respectively. We used the Mahalanobis distance to compute the trial-wise distances between the resulting MEG activity patterns, following the practice in [28]. We use a leave-one-trial-out cross-validation. Trials are sorted according to the orientation of the to-be-decoded item, for example, the current target: for each testing trial, all training trials were sorted into 10 groups according to the orientation in them relative to the orientation in the testing trial ([−90°, −72°], [−72°, −54°], [−54°, −36°], [−36°, −18°], [−18°, 0°], [0°, 18°], [18°, 36°], [36°, 54°], [54°, 72°], [72°, 90°]). For each time point, the Mahalanobis distance between the testing trial and each of the averaged activity patterns from the 10 groups was computed. The 10 resulting distances were mean-centered and the sign was reversed, and then their projections on the 0° vector corresponding to the tested orientation (i.e., each bin's distance was multiplied by the cosine of the bin orientation) were averaged to yield the decoding evidence. Higher decoding evidence corresponds to stronger and more successful decoding of the item. For the decoding of the previous target, we used this same method but took the target from the previous trial as the to-be-decoded item. The first trial from each block was also excluded from this analysis because there was no previous target.

Trials with too short (<0.65 s) or too long (>11.85 s) RTs were removed from the response-locked analysis because there is not enough analyzable data at the beginning or the end of the epoch. In the end, 366.95 ± 25.867 trials for each participant were included.

Cluster-based permutation tests with 100,000 iterations were conducted on decoding evidence to assess whether it differs significantly from 0.

## Estimation of neural bias

To test whether the representation of the current target was biased, we looked into the decoding results of the current target and took the mean Mahalanobis distances from three channels, [−72°, −54°], [−54°, −36°], [−36°, −18°], as clockwise evidence, and the mean Mahalanobis distances from another three channels, [18°, 36°], [36°, 54°], [54°, 72°], as counterclockwise evidence. The difference between them was calculated as the asymmetry index. A positive asymmetry index indicates a clockwise neural bias and a negative asymmetry index shows a counterclockwise bias. Trials were separated into two groups according to whether the inducer orientation (i.e., previous target or sample 1) was clockwise relative to the current target ("clockwise inducer trials", all trials where the inducer was [−90°, 0°] to the target) or counterclockwise relative to the current target ("counterclockwise inducer trials", all trials where the inducer was [0°, 90°] to the target). We compared the mean asymmetry index of the current target for these two groups of trials. If the asymmetry index for clockwise inducer trials is higher than the asymmetry index for counterclockwise inducer trials, the neural representation of the current target is biased attractively toward the inducer. If the asymmetry index for clockwise inducer trials is lower than the asymmetry index for counterclockwise inducer trials, the neural representation of current target is biased repulsively away from the inducer.

For the analyses on biased representation of the target caused by the participant's report (Fig 4B), we used the same method but took the participant's report as the inducer. Participants' reports are known to be subject to systematic biases that are independent of trial history, such as repulsive biases away from the cardinal and oblique axes [35, 44]. These context-independent biases in participants' reports were corrected first. Following the practice in [35], we fit a sinusoidal function (a sum of three sinusoids with possibly varying frequency, phase and amplitude) on participant's recall errors as a function of target orientation, and then removed the fit from the recall. Skipping this correction did not change the outcome of the analysis.

For all analyses on neural bias, cluster-based permutation tests with 100,000 iterations were also conducted on the asymmetry index from the two group of trials to assess whether there was a significant difference between them. Due to the higher level of noise in this data, none of the neural bias survived the cluster-based permutation test. Thus, we opted to testing the asymmetry indexes averaged in a time window with paired $t$ test.

### Searchlight analysis

To test which part of the brain contributed the most to the decoding or the neural bias, we also run searchlight decoding analyses. Iteratively for each one of the 306 sensors, we took this sensor and its 47 most closely adjacent sensors and run the decoding on this smaller set of data. By doing so, we can assess approximately which sensors were more important for the effects of interest. For the topographies of neural bias, the bias index was calculated by subtracting the asymmetry index of counterclockwise inducer trials from the asymmetry index of clockwise inducer trials. Thus, a positive bias index shows an attractive neural bias, and a negative bias index means a repulsive neural bias.

To compare the similarity between topographies, Pearson correlations were calculated.

### Replication in an independent EEG dataset

To replicate the findings from the MEG dataset, we conducted the same set of analyses on a previously acquired EEG dataset.

### Experimental procedure

Thirty participants performed two variants of a visual working memory task ('forced choice discrimination' and 'cued recall', see Fig 6A), in different blocks. The study was conducted in accordance with the Declaration of Helsinki and approved by the Central University Research Ethics Committee of the University of Oxford (MSD-IDREC-C1-2013-052). Because this reanalysis only used the data from the blocks of the cued recall, we describe the experiment procedure of the cued-recall task below.

The task to be performed (i.e., "forced choice discrimination" or "cued recall") was indicated at the beginning of each block via on-screen instructions. In the cued-recall task, participants were required to encode two sequentially presented, randomly oriented gratings in working memory, after which an auditory cue indicated which of the two items would be probed after a delay. The probe stimulus appeared at a random orientation, and participants dialed it to the orientation of the cued memory item.

Participants initiated each trial by pressing down the two response keys ("c" and "m" with their left and right index fingers, respectively) on the keyboard (and holding them down throughout the trial). The trial began with a fixation period (a black dot of 0.15° diameter was presented at the center of the screen on a gray [127,127,127] background throughout the trial) with a random duration (mean 1,000 ms, drawn from a truncated exponential distribution ranging from 750 to 2,000 ms), followed by the sequential presentation of two random noise stimuli (forward masks, presented at the center of screen 233 ms each, 250 ms minus one frame at a presentation rate of 60 Hz). Then, two randomly oriented Gabor gratings (6° diameter, 1.25 cycles/degree spatial frequency, 1.2° standard deviation of the Gaussian envelope, random phase, 50% Michelson contrast) were shown sequentially at the center of the screen. Each grating was presented for

233 ms (250 ms minus one frame), with a delay of 500 ms (i.e., item 2 appeared 750 ms after the onset of item 1). 500 ms after the offset of the second grating, a pure tone (100 ms, 440 or 880 Hz) cued which of the two items would be probed. The pitch of the tone (high or low) indicated the item (1 or 2). Tone mappings were counterbalanced across participants. After a memory delay (1,500 ms from onset of cue), a probe grating appeared with a random starting orientation. Participants were able to change the orientation of the probe stimulus (after an initial period of 300 ms during which its orientation was fixed) by releasing both keys and then pressing a key to dial the probe in the corresponding direction (left button for counterclockwise and right button for clockwise movement). The time when the two keys were released (relative to the time of probe onset) was recorded in each trial. When participants were satisfied that the probe was at the orientation of the cued memory item, they pressed the return key to submit their report. The response deadline was 15 s. After responding, feedback appeared after a delay of 50 ms (for 200 ms) in the form of two small light gray discs (0.25° diameter, gray) presented at the edge of the grating and indicating an imaginary line corresponding to the correct orientation of the cued item. Participants were instructed to respond as accurately as possible by minimizing the angular error between their response and the feedback orientation. The feedback presentation was followed by an inter-trial interval, which lasted until participants initiated the next trial.

During the delay between sample 1 and sample 2 ("Delay 1", 500 ms), the delay between sample 2 and the auditory retrocue onset ("Delay 2", 500 ms) and the delay after the retrocue onset and before the probe ("Delay 3′, 1,500 ms), two, two, and six noise stimuli (masks) were presented sequentially at 4 Hz (233 ms each, 250 ms minus one frame), respectively. Each noise stimulus (forward masks and masks) had the same size as the memory items and consisted of white noise convolved with a 2D Gaussian kernel (0.133° standard deviation, 50% contrast). Embedded within the noise stimuli were sub-threshold, randomly oriented Gabor gratings (20% contrast, random orientation and phase, 0.133 cycles/degree spatial frequency, 1.2° standard deviation). The embedded orientations were undetectable because of the overlaid noise. They were unrelated to the orientations of the two memory items on each trial.

Participants completed two sessions of the task. Each EEG session consisted of 12 experimental blocks (6 cued-recall blocks and 6 forced-choice-discrimination blocks, in pseudorandom order with the constraint that a single task could occur in no more than three consecutive blocks). There were 84 trials in each block, such that across both sessions participants completed 1,008 trials of each task type.

## EEG setup and preprocessing

The experiment was presented using Psychophysics Toolbox [45–47]. Visual stimuli were displayed on a Samsung Syncmaster monitor (1680X1050 resolution, 60 Hz refresh rate, screen width 47.5 cm, viewing distance 70 cm. EEG data were acquired using a Neuroscan Synamps 2 amplifier and Curry 7 software (Compumedics Neuroscan, Charlotte, NC) from 60 Ag/AgCl sintered surface electrodes, placed according to the 10-10 system. Data were recorded at 1,000 Hz. The anterior midline frontal electrode (AFz) was the ground electrode and all data were referenced online to the right mastoid (with left mastoid recorded for later re-referencing). Impedances were kept below 5 kΩ. Electrooculogram for blink and saccade monitoring was recorded from electrodes placed below and above the right eye and from electrodes placed near the outer canthi of each eye. EEG electrodes used were Fp1/z/2, AF7/3/z/4/8, F7/5/3/1/z/2/4/6/8, FT7/FC5/3/1/z/2/4/6/FT8, T7/C5/3/1/z/2/4/6/T8, TP7/CP5/3/1/z/2/4/6/TP8, P7/5/3/1/z/2/4/6/8, PO7/3/z/4/8, and O1/z/2.

Raw EEG data were imported into Matlab and preprocessed using EEGLAB [48] and Fieldtrip [49]. Data were re-referenced to the average of the two mastoids, downsampled to 250 Hz, bandpass-filtered between 0.1 and 45 Hz using an FIR filter, and epoched from 1,500 ms before onset of the first WM item in each trial to 3,000 ms after the onset of the probe. All trials were inspected visually for artifacts deriving from head motion, muscle artifacts, blinks, and saccades. Trials with artifacts were rejected. Noisy channels were replaced with a weighted average of neighboring channels using spherical interpolation. The remaining trials were further cleaned by removing artifacts related to eye blinks and saccades via independent component analysis. After converting the resulting data into Fieldtrip format, Fieldtrip's semi-automatic

rejection tool was used to reject any remaining trials with excessive variance. On average, 90.57% of trials were used in the final analysis.

After cleaning, the data were smoothed with an 8 ms Gaussian smoothing kernel. The mean across trials was removed from each time point and electrode. Trials were baseline-corrected (by regressing out the mean voltage between 200 and 50 ms before onset of the first item from each time point, separately for each sensor). Finally, the data were normalized to unit variance at each sensor and time point.

### Data analyses

We conducted all of our analyses on the data from the cued-recall blocks. For both behavioral and neural analyses, for the sample-1-induced-bias analyses, we used all trials where sample 2 was cued. For the previous-target-induced-bias analyses, we used all trials whose previous trial was not removed from the dataset (so that we have the information of the previous target orientation).

For the estimate of serial bias at the behavioral level, we use the same methods as in the MEG dataset.

We conducted multivariate decoding of the orientation of sample 1, sample 2, the current target, and the previous target separately. The decoding analyses were exactly the same as in the MEG dataset, with the exception that we did not exclude any trials due to a too-long or too-short RT. After PCA, across participants, 31.27 ± 12.26 principal components were retained. No response-locked decoding was conducted, because we do not have the data of when participants pressed the return key to submit their report in each trial, and the EEG epoch only has 3,000 ms of data after the probe onset, which may be too short for response-locked analyses.

The asymmetry indexes were calculated in exactly the same way as in the MEG analyses.

For statistical tests, we used the same cluster-based permutation tests and $t$ tests on data averaged in time windows, as in the MEG analyses.

## Supporting information

**S1 Text.  Supplementary analyses.**
(DOCX)

**S1 Fig.  The neural bias on the target induced by previous target (A), the decoding of the current target and the report orientation in the current trial (B), and the topographies of them in the time window where a marginally significant previous-target-induced attractive bias was found (C and D).** Yellow and purple horizontal lines in B indicate timepoints where a significant decoding of the target or the report orientation was found, respectively, as tested with cluster-based permutation tests. If the previous-target-induced attractive bias we found was merely driven by the rotated probe on the screen, we should be able to see a significant attractive bias in the time windows where the decoding of the report orientation is significant (purple-shaded regions in A). However, no significant attractive bias was found. In the only time window where a marginally significant previous-target-induced attractive bias was found, the topography of the attractive bias showed important contribution of the right central sensors (C) and the decoding of participant's report is most prominent in posterior sensors (D). Plotting conventions are the same as in Fig 2.
(TIF)

**S2 Fig.  The neural bias induced by the previous target (first row) and sample 1 (second row) during the encoding of the current stimulus.** (A) Bias of the neural representation of sample 1 caused by the previous target. Trials were sorted according to whether the previous target was clockwise (red line) or counterclockwise (blue line) to sample 1. The y-axis shows the mean asymmetry index, with a positive number indicating a clockwise-biased neural representation. Shadings around the thin lines indicate SEM. The gray-shaded regions indicate time windows when sample decoding was significant and over which data were averaged for statistical inference. (B) Bias of the neural representation of sample

2 caused by the previous target. Trials were sorted according to the orientation of the previous target relative to sample 2. (C) Bias of the neural representation of sample 2 caused by sample 1. Trials were sorted according to the orientation of sample 1 relative to sample 2. The black horizontal line indicates timepoints with a significant bias assessed with cluster-based permutation testing.
(TIF)

## Acknowledgments

We thank Eleanor Holt for assistance with data acquisition.

## Author contributions

**Conceptualization:** Jiangang Shan.

**Formal analysis:** Jiangang Shan.

**Methodology:** Jasper E. Hajonides.

**Project administration:** Nicholas E. Myers.

**Supervision:** Nicholas E. Myers.

**Visualization:** Jiangang Shan.

**Writing – original draft:** Jiangang Shan.

**Writing – review & editing:** Jiangang Shan, Jasper E. Hajonides, Nicholas E. Myers.

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
