## [Editor Report · Decision Letter 0]

6 Dec 2024

Dear Dr Shan, 

Thank you for submitting your manuscript entitled "Neural evidence for decision-making underlying attractive serial dependence" for consideration as a Research Article by PLOS Biology.

Your manuscript has now been evaluated by the PLOS Biology editorial staff as well as by an academic editor with relevant expertise and I am writing to let you know that we would like to send your submission out for external peer review.

Once your full submission is complete, your paper will undergo a series of checks in preparation for peer review. After your manuscript has passed the checks it will be sent out for review. To provide the metadata for your submission, please Login to Editorial Manager (https://www.editorialmanager.com/pbiology) within two working days, i.e. by Dec 08 2024 11:59PM.

Kind regards,

Christian

Christian Schnell, PhD

Senior Editor

PLOS Biology

cschnell@plos.org

---

## [Decision Letter · Decision Letter 1]

3 Feb 2025

Dear Dr Shan,

Thank you for your patience while your manuscript "Neural evidence for decision-making underlying attractive serial dependence" was peer-reviewed at PLOS Biology. It has now been evaluated by the PLOS Biology editors, an Academic Editor with relevant expertise, and by several independent reviewers. 

In light of the reviews, which you will find at the end of this email, we would like to invite you to revise the work to thoroughly address the reviewers' reports.

As you will see below, the reviewers think that the study is well executed and provides important insights. However, they also raise a number of concerns, many of which can be addressed with textual revisions to provide more explanations, tone down claims, and provide more careful interpretations, but some concerns may need additional data to be addressed.

Please note that Reviewer 3 was unable to submit a full report due to time constraints, but they have sent some comment which I have paraphrased below as coming from Reviewer 3. Let me know if you have any questions about those.

Given the extent of revision needed, we cannot make a decision about publication until we have seen the revised manuscript and your response to the reviewers' comments. Your revised manuscript is likely to be sent for further evaluation by all or a subset of the reviewers.

**IMPORTANT - SUBMITTING YOUR REVISION**

*Re-submission Checklist*

*Published Peer Review*

*PLOS Data Policy*

*Blot and Gel Data Policy*

Sincerely,

Christian

Christian Schnell, PhD

Senior Editor

PLOS Biology

cschnell@plos.org

REVIEWS:

Reviewer #1: The authors present a new analysis of data showing serial dependence in representations of orientation during recall. They also find a repulsive effect for ignored stimuli. The experiment adds to growing evidence for serial dependence in memory processes. The experiment is creative, the results are interesting, and the manuscript is well written. I do have a number of major concerns, listed below, and these can be readily addressed with revised and moderated text. 

1. The experiment design is more complicated than a traditional serial dependence experiment, which is fine, but it means some care is warranted in the interpretations. Unlike typical serial dependence (SD) experiments, stimuli are paired within single trials, along with added cues to selectively attend to one stimulus, and an attention task that encourages actively ignoring one sample in favor of the other. While some papers have used paired stimuli, and several serial dependence papers have studied the effect of attention, the design here is unique (in particular the added component of what some might consider active ignoring or suppressing). It's important to add caveats in the discussion that the complex pattern of results here may or may not extrapolate to all forms of serial dependence and may be a result of interactions between multiple mechanisms that involve different phenomena—more than just serial dependence. 

2. The feedback grating itself reinforces the target orientation. That could be an important source of a serial dependence signal. It's certainly an additional complication in the design that does not easily lend itself to simple interpretations. There is no response on that but it's highly correlated w target. Also highly correlated with response. And, no overt response is required for serial dependence to occur, as reported in many previous papers. 

3. Several papers have investigated the role of attention in serial dependence. Most of these find that attention boosts serial dependence. The experiment here is related to those studies, but it requires more than just selective attention. In some conditions here, observers must actively suppress or ignore, such as when the second sample was cued, requiring observers to ignore or suppress the first sample. Interestingly, Rafiei et al., 2020, APP, found that actively ignored stimuli caused a *repulsive* bias. That is essentially what is found here in this experiment as well, and it could explain the finding that "when the second sample was cued it was repulsively biased away from the first sample on the same trial" (p. 3). The Rafiei paper seems especially relevant to the current manuscript. 

4. More generally, the role of actively ignored things in serial dependence is under-studied and not well understood. This makes the manuscript more interesting, but it complicates interpretation and readers should be cautioned about this. The pattern of results could involve multiple mechanisms rather than the single encoding/decoding scheme proposed here. Because of the complicated design and possible involvement of multiple mechanisms, the conclusion that "attractive (but not repulsive) serial dependence arises during decision-making, and that priors that influence post- perceptual decision-making are updated by the previous trial's target, but not by other stimuli" may be an overgeneralized statement. It may only hold for this particular design.

5. The authors should tone down claims in abstract and discussion about this adjudicating the debate between perception and memory. The results don't address that directly, and the design is a memory-centric one (involving lots of delays, retention intervals, multiple competing stimuli in working memory, feedback stimuli, etc). The results show a positive SD for recalled targets and a positive bias in activity. this could be due to attention, rehearsal (plus attention), or something else in memory, but it doesn't rule out that SD can happen in perception too. An experiment that generates a perception effect (an effect on appearance ) is necessary for that. The authors could either run additional experiments using other (simpler, less memory-focused) designs that would strengthen their claims, or they can tone down the overly broad speculation about perception vs memory. 

6. What about the other sample? It seems like repulsion only reported for sample 1. What about the fellow stimulus (the ignored one, report 1st, sample 2)? Was there repulsion of that? If not , the story isn't as simple as implied. Referring readers to the previously published paper for analyses of each condition isn't sufficient here, if the current manuscript is intended to be a complete and independent published paper. If this manuscript is just a follow-up analysis of one particular condition then maybe it is better as a letter to the editor at J Neurosci? 

7. Results in Fig 4B and S1 cast doubt on the so-called response-based serial dependence, which has been reported in several papers, recently. Those papers claimed that serial dependence was the result of the report itself. The results here suggest otherwise. These results alone are very important to report, and should be highlighted in the discussion section as more than just a control, because they address an ongoing question in the literature. 

8. What are the decoding results for the feedback stimulus itself? Is it biased by the non-target inducer? Shouldn't it be, if the authors are correct about the repulsion effect? 

9. In Fig 2A, the right graph shows decoding of target after response. Presumably this is decoding the feedback stimulus, no? If not, then it's unclear what "response" really means here. Is it not what the methods say? "Responses were submitted by pressing another button." This "another button" defines the response time? And it's what counts as "response," no? It's not clear what to take away from the right panel of 2A. But it seems possible that there could be a bias in this decoding as well. 

10. "We did not find any bias of the neural representation of the current target caused by the previous target (Fig 2B left; t = 0.4816, p = 0.6356). However, a bias seemed to arise later in the recall period (Fig 2B left, starting from ~1000 ms)." There does appear to be a clear effect at ~1000msec. Directly comparing the left and right panels of 2B is not exactly fair because the stimuli are very different in those plots. When the probe onset happens, it is a salient static stimulus. That there isn't a bias in decoding does not mean there is no bias in the representation—it's a null result, and it could simply be due to the low SNR of the biasing effect. That is, the positive sequential effect might manifest before t=1000, but it's not measurable because the probe stimulus has a lot of orientation energy that swamps the biasing signal. As the observer starts adjusting the probe, the orientations presented on the screen have increasing variability/noise until the point that there is effectively a fully uniform distribution of orientations. In other words, moving the probe around in different ways on different trials adds noise and uncertainty that isn't present when the probe first appears. The authors sort of acknowledge this when they say that "the later part of the epoch contained increasing amounts of noise from irrelevant signals…" and they can test this themselves: plot stimulus orientations that the observers actually saw during the probe period. At time 0, there is no variance in orientation, because the random initial orientation of the probe is static at first. Once the observers start moving the probe around in different uncorrelated ways, there is increasing variability, though. That could add noise that actually makes the biased representation easier to detect. If the authors were to do another experiment where they just presented a white noise stimulus as the probe (rather than a high contrast, salient, noise free grating), they might find that the previous target orientation could be decoded at t = 100 msec, not just t = 1000 msec. We can't know without another experiment, but the point is that trying to directly compare the response aligned data (Fig 2B right) versus probe onset data (Fig 2B left) isn't quite fair. And, absence of proof is not proof of absence, and claims about this being evidence for "decision" processes that take time are tenuous. Instead, a more direct experiment is called for. 

11. Is accuracy related to the serial dependence effect? When observers were better at matching the target orientation, was there a stronger effect or more biased decoding? Is there a relationship between residual error (difference between reported orientation and feedback orientation) and sequential effect, biases measured or decoding bias?

12. P. 4. "The target orientation could be decoded from 254 to 834 ms after probe onset" Is the feedback stimulus a problem for this analysis? What is RT? It's reported in methods, I think, but not in the main text. According to Fig 1, 250 msec after probe onset is the feedback stimulus in which "grating with the correct orientation was presented as feedback." Add some indication of RT in Fig 1, if that's the reason the feedback stimulus doesn't influence this decoding analysis. Also address it in the manuscript to avoid reader confusion. 

13. P. 12 "trails" should be "trials"

14. What happened to the single grating data (half the data according to Fig 1)? What was the behavioral effect for that? 

15. The "reactivation" analyses are not entirely convincing because the stimuli on the screen are high contrast, noise-free, low uncertainty, etc, and could easily interfere or prevent measurement of reactivation even if it were present. Published papers reported that serial dependence is larger with more uncertain stimuli, in general, and so the reactivation may not be measurable not because it is absent but because it's washed out. Again, another example of arguing from null results. 

16. Not unrelated to the above, the stimuli in this experiment are not ideal for studying SD, if the goal is evaluating or making claims about SD in perception (as the discussion does). The gratings lack noise, apparently have high contrast, and have low uncertainty. Incidentally, the stimuli should be described in more detail: contrast, spatial frequency, size, etc. 

17. P. 9. "An alternative proposal is that serial dependence of working memory representations develops post-perceptually" The "perception" versus "memory" origin of serial dependence is not a black-and-white alternative. It is possible that serial dependence operates at many levels and in many processes including working memory AND in perception and in other domains (like long term memory, affect, motor control, decision making, and across modalities). Kiyonaga et al., 2017 acknowledged this first but it's worth repeating here.

18. P. 9. "our finding suggests different previously attended information may not be incorporated into the prior to the same degree" This is also the point that the Rafiei et al APP 2020 paper made. Rather than arguing for encoding/decoding difference (p 10), what if the key difference in the experiment here is ignoring versus attending? The timecourse of ignoring (non-target) is different than the timecourse of attending (target + feedback stimulus which reinforces the target) in this experiment, so it's not possible to use timecourse results or differences in timecourse to disentangle these alternatives. 

19. Discussion section needs to be revised and the claims about perception vs memory toned down. The "perception" vs "post-perception" debate is not directly addressed by these experiments, in part because of the concerns above, and in part because the experiment design was geared specifically to tax memory processes (see comments above about complexity of the design, competing stimuli, retention intervals, rehearsal, feedback stimulus, etc). It is true that the experiment says something positive about memory processes. But, simply put, this experiment does not get at perception directly. A new experiment is needed with, for example, a detection task like that in Murai et al., Curr Bio, 2021 or Weilnhammer, et al., Curr Bio, 2023, or a forced choice experiment like Cicchini, et al, JoV, 2017, or one like Collins, JoV, 2020, or another simpler/more direct design that takes advantage of the fact that SD (in perception, though maybe not in memory) is retinotopically specific (Collins, JoV, 2019). If new experiments are not added, then the discussion section should acknowledge these limitations and invite future work on the topic. 

Reviewer #2: The work here addresses a gap in knowledge about the time course and context of attractive and repulsive biases at the behavioral and neural level. Behavioral reports of orientation are often attracted towards the previously reported target. Moreover, in a visual working memory task with two oriented stimuli per trial (e.g., Hajonides et al. 2023 J Neuro), the orientation report of stimulus 2 is repulsively biased away from stimulus 1 and the previous target during encoding. 

On its surface, a repulsive neural bias from the previous target during encoding seems incongruent with an attractive behavioral bias. But as the authors state, there is a lack of direct investigation of neural bias at the post-perceptual decision-making stage, where this discrepancy might be resolved. This is of theoretical interest because the popular 'continuity field' model predicts that attractive biases are the result of integrating the current stimulus with the prior stimulus early in perceptual processing. Thus, observing a late, post-perceptual locus of attractive bias would run counter to the dominant model in the field. 

Here, the authors reanalyze data from a previous study (Hajonides et al. 2023 J Neuro), focusing on MEG data acquired during the memory-recall period (i.e., the decision-making stage) and make two key observations:

1. The neural representation of the current target is attracted to the previous target during the decision-making stage — consistent with behavioral reports being attracted to the previous target.

2. The neural representation of the current target is repelled from the other stimulus early in the decision making stage but is not attracted to the other stimulus from the current trial (i.e., there is only an attractive bias to the target on the previous trial). 

Thus, the authors find neural evidence for distinct early and late neural mechanisms for repulsive and attractive serial dependence during memory recall, respectively, and these observations are generally inconsistent with the continuity field model of serial dependence. 

First, the authors find that the current target orientation can be decoded 254 to 834 ms after the probe's onset, and thus they treat this period as a time window of interest. However, this was not the temporal epoch where they observed an attractive bias: a bias arose later, ~1000 ms after the probe onset, much closer to the average response time (Fig 2A left). The target orientation was also significantly decodable 163 to 0 ms before the subjects indicated their final response (Fig 2A right), and an attractive bias was found during this narrow time window, including the last 2000 ms before a response was made. 

Interestingly, during the early time window after the probe when the target was decodable, the authors observed a repulsive bias away from sample 1, consistent with the repulsion found in behavior. However, there was no bias induced by sample 1 in the response-locked analysis. 

Introduction

"Moreover, in our previous study [1], different items in trial history induced different behavioral effects: Participants' report in the current trial was attracted toward the cued item from the previous trial (henceforth "previous target"), while the report of the second sample was repulsively biased away from the first sample in the same trial ("sample 1"). However, during encoding the neural representation of the current stimulus was repulsively biased away from both sample 1 and the previous target." (pg. 2)

Due to wording here, it's not clear whether sample 2 is both attracted towards the previous target and repulsed from the current sample 1, but it seems like Fig 1B indicates that it is? 

Moreover, by this point in the manuscript, a reader might wonder whether sample 1 is biased by sample 2 in any way. Hajonides et al. 2023 J Neuro suggests no: "In trials with report first cues, there was no significant bias toward or away from the interfering second grating orientation that was not relevant to the task at hand (t(19)= 0.74, p = 0.467). In contrast, trials with report second cues revealed significant biases away from the initially encoded first grating orientation (t(19) = -2.33, p = 0.031; illustrated in Fig. 2B)."

"Our goals were two-fold: First, we wanted to provide neural evidence that the attractive bias arises during the post-perceptual decision-making stage. Second, we wanted to show that this attractive bias is context-dependent — namely that only the target from the previous trial, but not the other stimulus from the current trial, led to an attractive neural bias." (pg. 2)

Shouldn't these goals be rephrased as hypotheses? This would really help frame the importance of the questions being asked, especially if put in the context of prominent models like the 'continuity field'. While the discussion does a good job of putting these findings in context, the intro is narrowly focused. 

Results

"Target decoding was significant from 163 to 0 ms before the response (Fig 2A right; p<0.00001)."

Why was neural evidence significant for such a long time after the response, relative to the duration of significance after the probe onset?

Resiliency during memory-recall? Or some mechanism that is preventing contamination of the target memory during reconstruction?

"Fig 2. Decoding of the current target and the neural bias on the target induced by sample 1 or the previous target."

What biases would we expect to see if the trials were sorted by both the previous target and current sample 1? e.g., clockwise-previous-target-clockwise-sample-1 trials; clockwise-previous-target-counterclockwise-sample-1 trials. Do the biases interact?

Methods

"Due to the higher level of noise in this data, none of the neural bias survived the cluster-based permutation test. Thus, we opted to testing the asymmetry indexes averaged in a time window with paired t-test." 

In general, the effects are fairly subtle, and permutation testing is best practice. Do the authors have any other data sets (or can they find another open data set) that would partially replicate these findings? It is unlikely that there would be another paradigm with two stimuli on each trial, but there might be another study that at least has one target and a probe to assess probe-locked attractive biases? 

"Principal component analysis was applied on this vector to reduce its dimensionality, maintaining 90% of the variance."

A few questions: how do you do PCA on a vector? How many components, on average, were retained? Does the choice of 90% matter for the results? And why use Mahalanobis distance on data after PCA given that the covariance matrix of the components is a diagonal matrix with 0s for off-diagonal elements? In this case, Mahalanobis distance reduces to the normalized Euclidean distance. I don't think this matters to the results, but it does make me wonder if I correctly understand the analysis pipeline. 

"To test whether the representation of the current target was biased. We looked into the

decoding results of the current target and took the mean Mahalanobis distances from channels

[-18°, -72°] and [18°, 72°] as clockwise and counterclockwise evidence, respectively. The

difference between them was calculated as the asymmetry index."

What motivates the choice of these bins, especially the inclusion of channels 72 degrees away? Based on the behavioral data in this study, and many others, I would expect that attractive serial dependence would peak around 30-40 degrees, so I didn't follow why these non-consecutive bins were used. 

Minor Comments

The font in the figures is very small, and gets fuzzy when magnified. Not a big deal here, but just something to remedy moving forward. 

There were some minor typos:

"In the last 2000 ms before participants completed recall the previous target ..." (pg. 4)

"For the sample-1-induced bias, we focused on trial where …" (pg. 11)

"To analyzed the MEG activity in the probe period …" (pg. 11)

"We train the decoder on all non-rejected trials for the decoding of the current target." (pg. 11)

"Trails are sorted according to the orientation of the to-be-decoded item …" (pg. 12)

"To test whether the representation of the current target was biased. We looked into the decoding results of the current target …" (pg. 12)

"... the neural representation of current target is biased attractively toward the inducer, and vice versa." (pg. 12)

"... we use the same method but take the participant's report as the inducer." (pg. 12)

Reviewer 3:

Reviewer 3 noted that the paper is difficult to read and that important methodological details are missing, in particular when the authors refer to previous work without providing any details in the current manuscript. 

Furthermore, the indirect analyses and transformations on MEG data sometimes makes it challenging to interpret the findings. Reviewer 3's most important concern was that the metric used to measure similarity lacks a baseline. If the aim is to see a MEG signature of the serial bias, the authors need to demonstrate that the metric they choose in the first place reflects the target (without a bias).

---

## [Decision Letter · Decision Letter 2]

14 Jul 2025

Dear Jiangang,

Thank you for your patience while we considered your revised manuscript "Neural evidence for decision-making underlying attractive serial dependence" for publication as a Research Article at PLOS Biology. This revised version of your manuscript has been evaluated by the PLOS Biology editors, the Academic Editor and the original reviewers.

Based on the reviews and on our Academic Editor's assessment of your revision, we are likely to accept this manuscript for publication, provided you satisfactorily address the remaining points raised by the reviewers. Please also make sure to address the following data and other policy-related requests:

* We would like to suggest a different title to improve its accessibility for our broad audience: "Attractive serial dependence arises during decision-making". If you have an alternative suggestion, please reach out to me via email, so we can discuss the title before you re-submit your manuscript.

* Please add the links to the funding agencies in the Financial Disclosure statement in the manuscript details.

* Please include the approval/license number of the ethical approval for the experiments.

* Please include information in the Methods section whether the study has been conducted according to the principles expressed in the Declaration of Helsinki.

* DATA POLICY:

* CODE POLICY

* Please note that per journal policy, the model system/species studied should be clearly stated in the abstract of your manuscript. 

We expect to receive your revised manuscript within two weeks. 

*Published Peer Review History*

*Press*

Sincerely,

Christian

Christian Schnell, PhD

Senior Editor

cschnell@plos.org

PLOS Biology

Reviewer remarks:

Reviewer #1: 

The authors did a great job revising this paper. It is much stronger and more interesting in many ways. There's new data, new analysis, a more nuanced discussion section, among other improvements. I thank the authors for their effort.

I still have some concerns, but I think these can be considered "minor". These can be fixed with modifications to the text, mostly in the discussion section. With these changes to the text, I think the manuscript will make a great addition to the literature.

Fig 3 shows attractive effect pretty clearly in posterior regions. Aside from a control analysis that isn't related, the fact remains that the attractive effect seems stronger in posterior regions, no?. Even if not during encoding, the effect is in sensory areas? How do the authors interpret this? It's not as clear a rejection or confirmation of any particular hypothesis. Are the authors saying that there is attractive effect in decision making stage, which happens in posterior cortex? 

P11. "While this contrasts with the significant across-trial repulsive neural bias

reported in in our previous study [1] it is still inconsistent with the emergence of the attractive bias during encoding."

This is concerning for a couple of reasons. First, it does not replicate the other study findings. Second, it is a null result: it could be there is an attractive bias that has nulled the repulsive one. If we believe the first study results, then only way to null those in these data would be with an additional attractive bias. Of course any number of other possibilities exist. Point is that it's a null result not a rejection of any hypothesis or "inconsistent" with anything in particular. 

P 12. "Compared to the attractive neural bias in the MEG experiment (which first emerged at ~ 1000 ms after probe onset, Fig 2B), the attractive bias in the EEG experiment started earlier in the trial (after retrocue onset, Fig 8A)." This is another departure and again merits more balanced discussion. The picture is not nearly as clear as the authors would like. A takeaway from the eeg data is that attractive effects could be present earlier but undetected in the meg experiment for a variety of reasons. 

P14." Different from the results from the MEG analysis, in the EEG dataset we observed an attractive bias in

the recall period, with a significant difference detected by the cluster-based permutation test (Fig 9,

4108 to 4357 ms, p = 0.0409)."

Again, an important difference meriting nuanced discussion and not big broad claims about serial dependence only happening in one stage or another or only at one time point or another. The results are very interesting and informative, but they are just not that decisive. 

P 14 "but the attractive bias in the much shorter probe-onset-to-rotation-start

time window (see Fig 7A inset) failed to reach significance (t(29) = 1.6094, p = 0.1184"

This is clearly a decent effect size, it is positive, and it maybe even "moderate". Cohen's d nearly ~0.6? It doesn't reach the magic 0.05 p value level but it's a shorter time window, probably less statistical power, and perhaps the study is underpowered to actually address this question. It's clearly not a rejection, it's a positive trending result. An effect size that is "moderate" can't be ignored. It suggests there may be an attractive bias on the stimulus when it first appears, and it begs more experiments, more data. How is that effect size consistent with the late decision making story?

To be clear, having an effect this early would not disprove a later effect, but the later effect does not disprove the possibility of an earlier one either. 

In light of the above, add some discussion material about the mixed results above and trending effects and how these might impact the the debates in the literature. For example, one place to acknowledge the results above might be in the discussion section paragraph starting with "While the current results speak for the two-stage models in which the attractive serial bias develops post-perceptually in a working memory task, they do not rule out the possibility that the attractive serial bias can emerge at the perceptual level". 

P 15 "This rules out that the

observed attractive neural bias was driven by a reactivation of the previous target during decision-

making."I think some readers might have concerns with the strength of this conclusion, because the reactivation signal could simply be too small to detect, or swamped. I think the authors acknowledged this well in the discussion section but the strong claim is made in the result section. So, the result section needs the acknowledgment too.

Discussion. Sets up hypothesis A (one stage) vs hypothesis B (two stage). It's simple but the results aren't. Consider adding a third hypothesis C: both. The results are simply mixed (search for the word "discrepancy") and the two hypotheses are not actually mutually exclusive. It could be that repulsive and attractive effects happen at each stage? Or some other hybrid? Can the authors take this opportunity to speculate a bit about next-level accounts, future hypotheses, rather than trying to shoehorn clear differences in results into conforming to hypothesis B?

Discussion " however, we did not see a

bias induced by the previous target during the current-trial encoding of either sample 1 or sample 2. This discrepancy could be explained by…" …Lots of things. One possibility is that attractive and repulsive effects are both present early but often dominated by repulsive ones simply bc of adaptation strength. That doesn't rule out the possibility of an attractive effect hiding in there and perhaps the discrepancy in the two studies is the sensitivity to the attractive effect. 

Title: in light of the results I think the title should change the word "underlying" to "contributions"

Reviewer #2: Great job with the revision, and the inclusion of the new EEG data is a real strength.

---

## [Editor Report · Decision Letter 3]

23 Jul 2025

Dear Jiangang,

Thank you for the submission of your revised Research Article "Attractive serial dependence arises during decision-making" for publication in PLOS Biology. 

I have received your out-of-office note in response to my previous email. Since it is a relatively minor change, I have just gone ahead and made this tweak to the abstract (adding "human" in front of "magnetoencephalography"). Please reach out to me or and the production team in the subsequent stages if you have any concerns about this change. 

Otherwise, on behalf of my colleagues and the Academic Editor, Thorsten Kahnt, I am pleased to say that we can in principle accept your manuscript for publication, provided you address any remaining formatting and reporting issues. These will be detailed in an email you should receive within 2-3 business days from our colleagues in the journal operations team; no action is required from you until then. Please note that we will not be able to formally accept your manuscript and schedule it for publication until you have completed any requested changes.

PRESS

Sincerely, 

Christian

Christian Schnell, PhD

Senior Editor

PLOS Biology

cschnell@plos.org